



# On the importance of snowpack stability, its frequency distribution, and avalanche size in assessing the avalanche danger level: a data-driven approach

Frank Techel[1,2], Karsten Müller[3], and Jürg Schweizer[1]

[1]WSL Institute for Snow and Avalanche Research SLF, Davos, Switzerland
[2]University of Zurich, Department of Geography, Zurich, Switzerland
[3]Norwegian Water Resources and Energy Directorate NVE, Oslo

*Correspondence to:* Frank Techel (techel@slf.ch)

**Abstract.** Consistency in assigning an avalanche danger level when forecasting or locally assessing avalanche hazard is essential, but challenging to achieve, as relevant information is often scarce and must be interpreted in the light of uncertainties. Furthermore, the definitions of the danger levels, an ordinal variable, are vague and leave room for inter-pretation. Decision tools, developed to assist in assigning a danger level, are primarily experience-based due to a lack
of data. Here, we address this lack of quantitative evidence by exploring a large data set of stability tests (N = 10,125) and avalanche observations (N = 39,017) from two countries related to the three key factors that characterize avalanche danger: snowpack stability, its frequency distribution and avalanche size. We show that the frequency of the most un-stable locations increases with increasing danger level. However, a similarly clear relation between avalanche size and danger level was not found. Only for the higher danger levels the size of the largest avalanche per day and warning region
increased. Furthermore, we derive stability distributions typical for the danger levels 1-Low to 4-High using four stability classes (*very poor*, *poor*, *fair* and *good*), and define frequency classes (*none or nearly none*, *a few*, *several* and *many*) describing the frequency of the most unstable locations. Combining snowpack stability, its frequency and avalanche size in a simulation experiment, typical descriptions for the four danger levels are obtained. Finally, using the simulated snow-pack distributions together with the largest avalanche size in a step-wise approach, as proposed in the Conceptual Model
of Avalanche Hazard, we present an example for a data-driven look-up table for avalanche danger assessment. Our findings may aid in refining the definitions of the avalanche danger scale and in fostering its consistent usage.

## 1 Introduction

Consistent communication of regional avalanche hazard in publicly available avalanche forecast products is paramount to avoid misinterpretations by the users (Techel et al., 2018). A key information in public bulletins is the avalanche danger
level. The danger levels - from 1-Low to 5-Very High - are described in the European Avalanche Danger Scale (EADS, EAWS, 2018) or its North American equivalent, the North American Avalanche Danger Scale (e.g. Statham et al., 2010)




with brief definitions of the key factors. The key factors that characterize avalanche danger are (Meister, 1995; EAWS, 2020, 2018):

– the probability of avalanche release,

– the frequency and location of the triggering spots, and

– the expected avalanche size.

These elements are expected to increase with increasing danger level (e.g. Schweizer et al., 2020).

The release (or triggering) probability of an avalanche, or 'sensitivity to triggers' as termed in the Conceptual Model of Avalanche Hazard (CMAH, Statham et al., 2018a), is inversely related to snowpack stability, with a higher probability for an avalanche to release with lower stability, and vice versa (e.g. Föhn and Schweizer, 1995; Meister, 1995). Hence, the

probability of avalanche release refers to a specific location and relates to the local (or point) snow instability. The latter has recently been revisited and three elements were suggested to describe point snow instability: failure initiation, crack propagation and slab tensile support (Reuter and Schweizer, 2018).

The actual spatial distribution of snow stability is typically unknown. So far, it can only be assessed with laborious extensive sampling (e.g. Birkeland, 2001; Reuter et al., 2016). However, in a regional avalanche forecast the spatial distribution

of snow instability can be described with regard to the frequency and the locations of triggering spots (or more generally the locations where the snowpack is weakest). From these two components, frequency and location, only frequency is relevant when assessing the danger level. The frequency always refers to a specific area, typically a forecast region. In addition, in the forecast the slope aspects and elevation are described where the danger prevails. The frequency describes the question «How often do spots with a certain snow stability exist within a region?» – in terms of numbers,

proportions or percentages. Typical frequency distributions for the danger levels 1-Low to 3-Considerable were described by Schweizer et al. (2003) using five classes of snow stability. In contrast, the location of triggering spots or of snowpack stability describes «Where in the terrain is avalanche release most likely?» Currently, the frequency is described using the terms *single*, *some*, *many*, and *most* (**?**), while terms describing the location are manifold (e.g. *where the snowpack is shallow*, *close to ridgelines*, *in bowls*, ...). In contrast, in the CMAH, the spatial distribution is related to the ease of finding

evidence of an avalanche problem (Statham et al., 2018a), and is described using the three terms *isolated*, *specific* and *widespread*.

Finally, avalanche size is defined with sizes ranging from 1 to 5 relating to the destructive potential of an avalanche (e.g. CAA, 2014; EAWS, 2019; McClung and Schaerer, 1981).

The EADS descriptions of the key factors for each of the five categories of danger level leave ample room for interpre-

tation and are even partly ambiguous. This may be a major reason for inconsistencies noted in the use of the danger levels between individual forecasters or field observers, and even more prominent between different forecast centers and avalanche warning services (Lazar et al., 2016; Statham et al., 2018b; Techel and Schweizer, 2017; Techel et al., 2018), but also when assessing different avalanche problems (Clark and Haegeli, 2018).





The same danger level can be described with different combinations of the three factors. To improve consistency in the use of the danger levels, a first decision aid, the Bavarian Matrix was adopted by EAWS in 2005. The Bavarian Matrix, a look-up table, combined the frequency of triggering locations with the release probability. In 2017, an update of the Bavarian matrix, now called the EAWS-Matrix, was presented that additionally incorporates avalanche size (EAWS, 2020).

More recently, a so-called Avalanche Danger Assessment Matrix (ADAM, Müller et al., 2016) was proposed, which tries to combine the workflow described in the CMAH with the assignment of the danger levels based on the three factors as suggested in the EAWS-Matrix. Both, the current version of the EAWS-Matrix and ADAM, are work in progress.

Challenges in the improvement of these decision support tools include the fact that the three key factors characterizing avalanche danger are not clearly defined and hence poorly quantified (Schweizer et al., 2020). Our objective is therefore

to address this lack of quantitative evidence by exploring observational data relating to snowpack stability, its frequency distribution and avalanche size. The data originate from different snow climates, but also from different avalanche warning services (Norway, Switzerland). The key questions are: (1) How do the three factors relate to the danger levels? And (2) Which combination of the actual value of the three factors does best describe the various danger levels? We present a methodology to generate data-driven stability distributions and to obtain class intervals describing the frequency of a

given snow stability class. Finally, we will compare the findings with currently used definitions in avalanche forecasting, as EADS and CMAH, and make recommendations for improvements towards more consistent usage of the danger scale.

## 2 Data

All the data described below were recorded for the purpose of operational avalanche forecasting in Norway (NOR; Norwegian Water Resources and Energy Directorate NVE) or Switzerland (SWI; WSL Institute for Snow and Avalanche

Research SLF). In the vast majority, these observations were provided by specifically trained observers, belonging to the observer network of either the Norwegian or the Swiss avalanche warning service.

Despite the necessity to harmonize some of the data across warning services, we argue that making use of data from different warning services and snow climates, may highlight potential biases' or differences.

### 2.1 Avalanche danger level

The target variable, the avalanche danger level, we want to describe the three factors with, is an estimate at best, as there is no straightforward operational verification. Whether assessing the danger level in the field or in hindsight, it remains an expert assessment (Föhn and Schweizer, 1995; Techel and Schweizer, 2017).

We rely on the local danger level estimates provided by specifically trained observers. In both countries, this estimate is based on the observations made on the day and on other information considered relevant (Kosberg et al., 2013; Techel

and Schweizer, 2017) and can be called a local nowcast. In very few exceptions (19 days during the verification campaigns in the winters 2002 and 2003 in the region surrounding Davos, SWI) a «verified» regional danger rating was available (Schweizer et al., 2003; Schweizer, 2007b).





**Table 1.** Data overview.

| parameter | | country | N | data from* |
|---|---|---|---|---|
| avalanches | natural | SWI | 29,511 | 2001-2002 |
| | human-triggered | SWI | 3,751 | 2001-2002 |
| | natural | NOR | 4,555 | 2014-2015 |
| | human-triggered | NOR | 1,200 | 2014-2015 |
| RB | | SWI | 4,698 | 2001-2002 |
| ECT | | SWI | 2,745 | 2007-2008 |
| | | NOR | 2,682 | 2014-2015 |

\* - for days between (and including) 1 Dec and 30 Apr. Always until (and including) the
winter 2018/2019.

In this study, we make use of local estimates for dry-snow conditions only. Each stability test or avalanche observation
was linked to a danger rating as described below (Sect. 2.2 and 2.3). If no local danger level estimates were available,
the data were not used.

Throughout this manuscript, we refer to the danger levels using their integer-signal word combination, e.g. 1-Low or 2-
Moderate.

## 2.2 Snow stability

Operationally available information directly related to snow instability includes simple field observations as well as snow
stability tests (Schweizer and Jamieson, 2010). Field observations such as recent avalanching, shooting cracks and
whumpfs (a sound audible when a weak layer fails due to localized loading) clearly indicate snow instability (Jamieson
et al., 2009; Schweizer and Jamieson, 2010). These observations are often made in the backcountry while ski touring
and do not require a person to dig a snow pit. Snow stability tests, on the other hand, are considered targeted sampling
(McClung and Schaerer, 2006) with the aim to assess point snow instability. Here, we used data obtained with two stability
tests regularly used to assess snow instability in Switzerland and Norway, the Rutschblock test and the Extended Column
Test.

The **Rutschblock test (RB)** is a stability test, ideally performed on slopes steeper than 30°, where a 1.5 m × 2 m block of
snow is isolated from the surrounding snowpack and loaded by a person (e.g Schweizer, 2002). An observer performing
a RB records which of the 6 loading steps, referred to as the *score*, caused failure, and what portion of the block slid (the
*release type*: whole block, most of block, edge only). If no failure occurs, RB7 is recorded. *Score* and *release type* pro-
vide information on failure initiation and crack propagation, essential components of a slab avalanche release (Schweizer
et al., 2008b). RB data were only available from Switzerland.

The **Extended Column Test (ECT)** is a stability test that provides an indication on crack propagation propensity (Simen-





hois and Birkeland, 2006, 2009). In contrast to the RB, the ECT is performed on a comparably small (30 cm × 90 cm) isolated column of snow and loaded by tapping on the block. The observer records the tap at which a crack initiates (1-30) and whether a fracture propagates across the entire column (ECTP), or not (ECTN; Simenhois and Birkeland, 2009). If no fracture is initiated with 30 taps ECTX is recorded.

Each stability test was linked to a danger rating relating to dry-snow conditions. We considered the danger rating most relevant, which was transmitted together with the snow profile or stability test (in text form, SWI). In the Swiss data set, this danger rating was replaced for stability tests observed on days and in warning regions, for which a «verified» regional danger rating existed (Sect. 2.1). If neither of them was available, the operational database was searched for local danger level estimates reported during the day and in the same region. Often, these local estimates were reported by the same

observer who performed the test.

The Swiss RB data set comprised 4,698 RBs, observed mainly on NW-, N-, and NE-facing slopes (67%) at a median elevation of 2,380 m a.s.l. (interquartile range IQR 2,160-2,565 m) and a median slope angle of 35° (IQR: 32-37°). The Swiss ECT data set contained 2,745 ECTs; 67% were observed in NW-, N- and NE-facing slopes at a median elevation of 2,372 m a.sl. (IQR 2,134-2,547 m) and at 34° (IQR 31-36°). The Norwegian ECT data set consisted of 2,682 ECTs,

observed at a median elevation of 760 m a.sl. (IQR 730-1,067 m). Consistent information on the slope aspect was not available for Norwegian stability data.

## 2.3   Avalanches

As part of the daily observations, observers (and occasionally the public) reported avalanches observed in their region. Avalanches can be reported individually, but also by summarizing several avalanches into one observation. While in-

dividual avalanches were reported in a similar way in NOR and SWI, the reporting of several avalanches differed. In NOR, observers reported avalanche size, trigger type and wetness, which was typical for the situation, and described the observed number of avalanches using categorical terms (single: 1, some: 2-4, many: 5-10, numerous: ≥11). In SWI, observers reported the number of avalanches of a given size. In all reporting forms, information about the wetness and trigger type could be provided. In either country, avalanche size was estimated according to the destructive potential, and

a combination of total length and volume, resulting in avalanche sizes of 1 to 5 (EAWS, 2019). In SWI until 2011, only size classes 1-4 were used.

The analysis was restricted to dry-snow avalanches, where the trigger type was either natural release or human-triggered. These avalanches were linked to a dry-snow local danger rating for the release date of the avalanche(s) and in the same warning region.

To enhance the quality of the data, we filtered observations, which we believe may indicate errors in the local estimate of the danger level or of avalanche size. To this end, we calculated the avalanche activity index (AAI, Schweizer et al., 1998), a dimensionless index summing up avalanches according to their size with weights of 0.01, 0.1, 1, and 10 for avalanche sizes 1 to 4, respectively. We did not assign weights to the trigger type (natural, human-triggered). For NOR, where the number of observed avalanches is described categorically, we assigned numbers as follows: one = 1, few (2-5)



= 3, several (6-10) = 8, numerous (≥11) = 12. For each country, we then rank-ordered the avalanche data and the lowest 2.5% of the days and regions with 2-Moderate, 3-Considerable and 4-High, and the top 2.5% of the days and regions with 1-Low, 2-Moderate or 3-Considerable were removed.

The total number of avalanches that remained was 5,755 in Norway, observed on 1,618 different days and regions, and

33,262 in Switzerland, observed on 6,610 days and regions (Table 1).

## 3  Methods

### 3.1  Classification of snow stability

In the following, we describe how we classified the results of the snow instability tests in the four stability classes (*very poor*, *poor*, *fair* and *good* - stability class names are in italics throughout this manuscript).

**Rutschblock test (RB)** results were classified into the four stability classes according to Figure 1a using a combination of score and release type, which have been shown to be good predictors of unstable conditions (e.g. Föhn, 1987; Jamieson and Johnston, 1995; Schweizer et al., 2008b). This stability rating is close to the operationally applied stability rating in Switzerland, which includes five classes and in addition considers weak layer properties and snowpack structure (Schweizer, 2007a; Schweizer and Wiesinger, 2001). The classification by Schweizer (2007a) was used in Techel and

Pielmeier (2014) for an automatic assignment of stability based on RB score and release type (also five classes). As in Techel. et al. (in prep.), we combined the two classes *very good* and *good* into one class called *good*.

Recently, a similar classification was proposed for the **Extended Column Test (ECT)** (Techel. et al., in prep.). Using a combination of crack propagation and the number of taps until failure initiation, four stability classes were defined (Fig. 1b). Techel. et al. (in prep.) compared the RB- and ECT-classifications shown in Fig. 1 to slope stability classified as either

unstable or stable. They showed that with increasing stability class, the proportion of slopes rated as unstable decreased. In a data set with 30% unstable and 70% stable slopes, the four RB stability classes included 76%, 53%, 25% and 11% unstable slopes, while the four ECT stability classes included 57%, 40%, 23% and 15% unstable slopes. This indicates that the four stability classes for RB and ECT do not exactly line up: The second RB class had a proportion unstable slopes (53%) similar to the first ECT class (57%), and the second ECT class (40%) had a value in-between the second

and third RB classes (53% and 25%). To accommodate this misalignment, we assigned the following four class labels to the four ECT classes: *poor*, *poor-fair*, *fair* and *good*. It is of note, that ECT class *poor* also includes the weakest ECT results, which may be associated with *very poor* stability. To obtain the lowest RB or ECT stability class at each location, we proceeded as follows: If the depth of a weak layer failure was less than 5 cm below the snow surface, we considered the failure as not relevant (rating the test result as *good*), if a failure layer was between 6 and 10 cm below the snow

surface, we increased the stability rating by one step (e.g. from *very poor* to *poor*). If several failure planes were detected in a single stability test, the most unstable stability class was retained for further analysis.





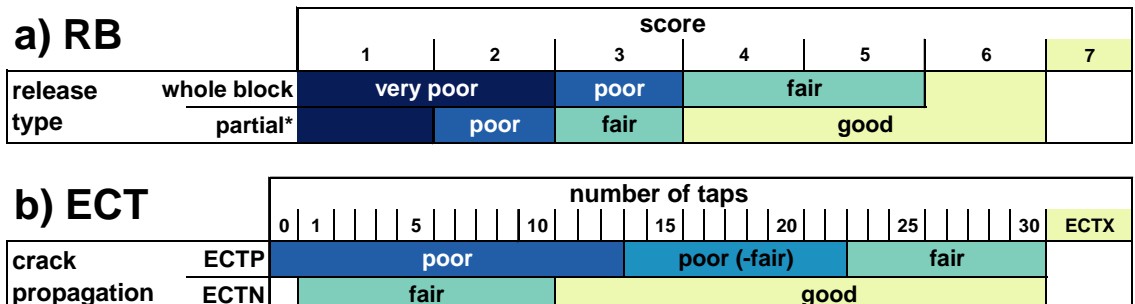

**Figure 1.** Stability classification of (a) Rutschblock test results (based on Schweizer (2007a); Techel and Pielmeier (2014)) and (b) Extended Column Test results (based on Techel. et al. (in prep.)). * - part of block includes release types most of block and edge only

### 3.2 Simulation of snow stability distributions

To determine the distribution of point snow instability within a defined region and at a given danger level many stability test results on a given day are in general needed (e.g. Schweizer et al., 2003). However, as we most often only had one stability test result on a given day, we followed an alternative approach. Assuming that a single test result is just one

sample from the stability distribution on that day and that different days with the same danger level exhibit similar stability distributions, we generated stability distributions by random sampling from the entire population of stability tests at a given danger level. Thus, we applied bootstrap sampling (Efron, 1979) and proceeded as follows (see also Fig. 2, steps 1 and 2):

- – (i) We randomly selected $n$ stability test results with replacement from the stability tests associated with the same
danger level, resulting in a single bootstrap sample. We repeated this procedure $B$ times for each danger level.

- – (ii) For each of the $B$ bootstrap samples, we calculated the proportions of *very poor*, *poor*, *fair* and *good* stability tests.

Bootstrap sampling, frequently used to estimate the accuracy of a desired statistic or for machine learning (Hastie et al., 2009), requires a sufficiently large number of replications $B$ to be drawn. We used $B = 2,500$ for each danger level,
resulting in 10,000 stability distributions in total.

The second important parameter when bootstrap sampling is the number $n$ of stability tests drawn in each sample. Small values of $n$ increase variance, and hence overlap between samples drawn from different danger levels, and reduce the resolution of the desired statistic (e.g. for $n = 10$, the resolution is 0.1, for $n = 100$ it is 0.01). We wanted not only some overlap between distributions sampled from different danger levels - Nature is not as discrete as the danger levels may
suggest, but also a preferably high resolution of our statistic. Unfortunately, there are no studies we can refer to concerning the amount of overlap that would be appropriate. We tested $n=\{10, 25, 50, 100, 200, 1,000\}$.





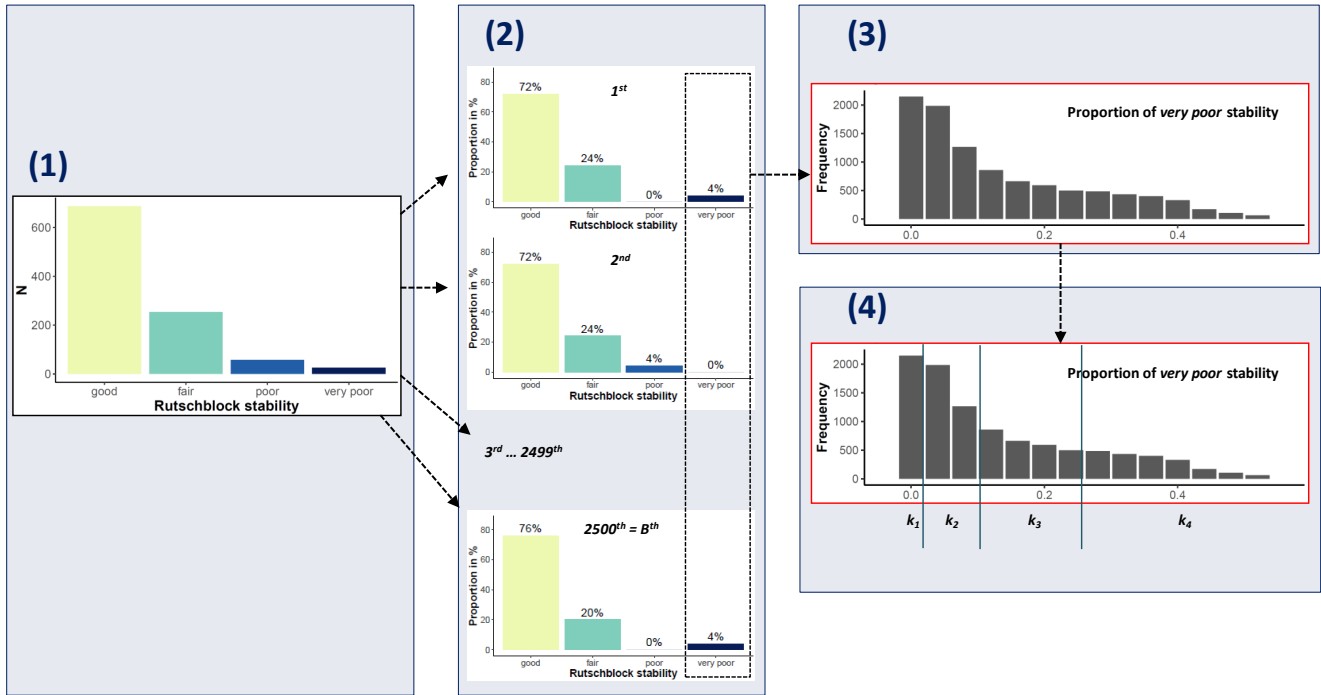

**Figure 2.** Schematic representation of the workflow for bootstrap sampling and frequency class definition. 1 - For each danger level, all stability tests are combined. 2 - From the combined stability tests (1), *n* tests are randomly sampled. This is repeated *B* = 2,500 times to obtain typical stability distributions for each of the four danger levels. 3 - The 4x 2,500 boot-strap samples are merged and the proportion of *very poor* rated stability tests per sample is plotted as a histogram. 4 - The statistics required for frequency class definitions are calculated and the *k* frequency classes defined. For details refer to the description in the Sections 3.2 and 3.3.

These simulations are compared to a small number of days when more than 6 RB tests (N=41) or more than 6 ECT tests (N=31) were collected in the surrounding of Davos (SWI).

### 3.3 Snow stability and its frequency - approach to define frequency classes

Currently, no classification exists that provides thresholds for the frequency a certain snow stability class is present. In
the following, we therefore introduce a data-driven approach to define class intervals that we will use to describe the frequency of a certain snow stability class. We considered the following points:

  – Classes should be defined based on the snow stability class most relevant with regard to avalanche release, hence the frequency of the class *very poor*. Even though the focus is on the proportion of *very poor* snow stability, classes need to capture the entire possible parameter space, i.e. from very rare to virtually all (1 to 99%).

– The number of classes should reflect the human capacity to distinguish between them. We explored 3, 4 and 5 classes only, as these are the number of classes currently used to describe and communicate avalanche hazard





and its components (e.g. three spatial distribution categories in the CMAH, four frequency terms in the EAWS matrix, five danger levels, five avalanche size classes).

 – Classes must be sufficiently different to ease classification by the forecaster as well as communication to the user. And, if quantifier terms were assigned to these classes, these terms would need to unambiguously describe such

increasing frequencies. An example of such a succession of five terms is *nearly none*, *a few*, *several*, *many* and *nearly all* (e.g. Díaz-Hermida and Bugarín, 2010).

Data-driven approaches for defining interval classes are numerous, and are described for instance for thematic mapping (e.g. Slocum et al., 2005) or for selecting histogram bin-widths (e.g. Evans, 1977; Wand, 1997). In general, the choice of class intervals should be appropriate to the observed data distribution. Approaches include, among others, splitting the

parameter space into equal intervals, into intervals with an equal number of observations in each bin, or finding natural breaks in the data by minimizing the within-class variance while maximising the distance between the class centers (e.g. Fisher-Jenks algorithm, Slocum et al., 2005). However, in our case, in which low values of the proportion of *very poor* stability are frequent and higher values rare, we made use of a geometric progression of class widths, considered most suitable for this type of distribution (Evans, 1977). Using this approach, we classified the data into *k* classes with class

interval limits being {0, $a$, $ab$, $ab^2$, ..., $ab^{k-1}$, 1}, where $a$ is the size (width) of the initial (lowest) class and $b$ is a multiplying factor. According to Evans (1977), a data-driven calculation of $b$ for the closed interval from 0 to 100 can be given:

$$b = \left( \frac{100 - VP_{\mathrm{med}}}{VP_{\mathrm{med}}} \right)^{\frac{2}{k}},$$
(1)

where *$VP_{med}$* is the median proportion of *very poor* stability, and *k* the number of classes preferred. This approach requires a suitable value of the number of classes *k* to be defined. Given *k* and *b*, the initial class width *a* is (Evans,

1977):

$$a = \frac{VP_{\mathrm{med}}(1-b)}{1 - b^{\frac{k}{2}}}$$
(2)

To derive $a$ and $b$, we generated snow stability distributions, as outlined in the previous section (see also Fig. 2, steps 3 and 4).

**3.4  Combining snow stability and its frequency with avalanche size: a simulation experiment**

When assigning a danger level, the information relating to snow stability and its frequency needs to be combined with avalanche size. As we cannot link these three factors using data relating to the same day and region, we used a simulation approach by assuming that the distribution of the observed data represents the typical values and ranges at a specific danger level. Randomly sampling and combining a sufficient number of data points results in typical combinations of the

three factors according to their presence in the data, but may also produce a small number of less likely combinations. We made use of the simulated frequency distributions of snow stability and their respective frequency class (Sect.s 3.2,





3.3). For each danger level, we complemented the snow stability information with avalanche size by randomly selecting an avalanche size from the empirical avalanche size distribution for the given danger level (which will be shown in Sect. 4.2) .

## 4  Results

### 4.1  Snow stability

#### 4.1.1  Rutschblock and ECT stability

We analyzed the distribution of RB and ECT results at danger levels 1-Low to 4-High (Fig. 3).

The proportion of *very poor* rated RB tests increased monotonically with increasing danger level from 2% at 1-Low to 38% at 4-High. As a consequence, the combined proportion of *very poor* and *poor* rated tests also increased strongly from 8% to 67% (Fig. 3a), while the proportion of tests rated as *good* decreased accordingly (68% to 10%). These patterns were also confirmed when exploring the correlation between the RB stability class and danger level (Spearman rank-order correlation; $\rho = 0.4$, p $< 0.001$).

The proportion *poor* rated ECT increased from 11% at 1-Low to 28% at 3-Considerable, while the proportion of the two most unfavorable stability classes combined rose from 19% to 44%. At 4-High only the combined proportion of the two most unfavorable classes showed this increasing trend (61%, Fig. 3b). Again, a positive though weak correlation between stability rating and danger level was noted (ECT: $\rho = 0.22$, p $< 0.001$). ECTs were conducted more frequently at higher danger levels in Switzerland than in Norway (e.g. at 3-Considerable: 39% in SWI and 21% in NOR). The ECT stability class distributions for the two countries are shown in in the Appendix (Fig. A1).

In both countries, very few RB and ECT were observed at 4-High (for instance ECT in NOR N = 6, in SWI N = 7, see also Fig. A1 in Appendix).

#### 4.1.2  Simulated stability distributions and frequency classification

As shown in the previous section, the RB stability classes *very poor* and *good* correlated better with the four danger levels than the ECT. For this reason, and because ECT seems not to separate well between *very poor* and *poor* stability, in the following we present results for RB only. The respective analysis for the ECT is shown as a supplement in the Appendix (Sect. 1).

To obtain a variety of frequency distributions of point snow instability, we sampled stability tests as described in Sect. 3.2. As outlined there, one important parameter affecting such a sampling approach is the number of tests *n* drawn in each sample. We tested *n* = {10, 25, 50, 100, 200, 1000}. We visually checked the resulting histograms for the proportion of *very poor* stability (Figure B1a-f in the Appendix) and visually checked for clusters in a two-dimensional context by considering the two extreme classes of the stability range, the proportion of *very poor* and *good* tests (Fig. C1).



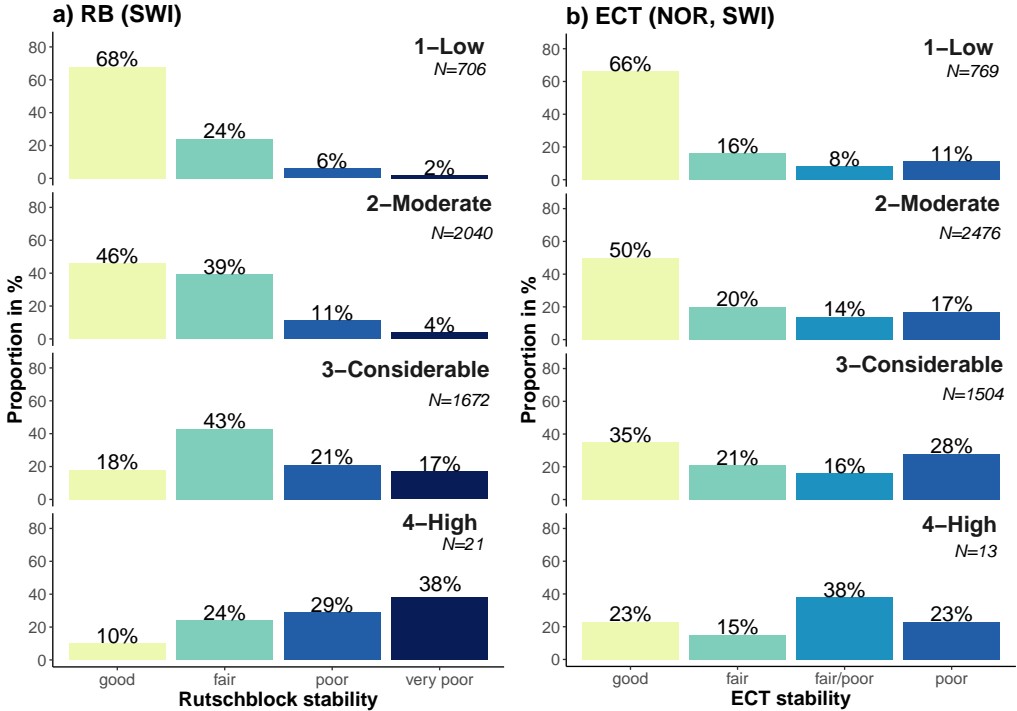

**Figure 3.** Distribution of stability ratings for the stability tests (a) Rutschblock (RB) and (b) ECT for danger levels 1-Low to 4-High. For the definition of the stability classes refer to Fig. 1 and Sect. 3.1.

The distribution of the proportion of *very poor* stability was skewed towards lower proportions being more frequent than higher proportions (Figure B1a-f). Increasing *n* impacted the number of modes detected in the histograms, with two or more modes being present when *n* reached values of about 50. This decrease of variance with increasing *n*, which leads to less overlap in samples drawn from different danger levels, is a characteristic of bootstrap sampling. Similar patterns

can be noted in the two-dimensional context (Fig. C1), with clusters not only becoming visually more and more pronounced with increasing *n*, but the overlap between danger levels reducing particularly at 3-Considerable and 4-High. Comparing the bootstrap-sampled distributions with actually observed distributions of stability tests on the same day and in the same region (N = 41), showed that the distribution obtained using bootstrap-sampling reflected the variation in the observed distributions reasonably well (Fig. 4). The influence of a low number *n* of tests drawn in the bootstrap or tests

actually collected in the field, is reflected in the large overlap between danger levels, but also variation within.

Relevant parameters for the definition of class intervals, as introduced in Sect. 3.3, are the respective median proportion of *very poor* stability $VP_{med}$ and the number of classes *k* desired. $VP_{med}$ showed a minor decrease with increasing resolution of the test statistic defined by *n*. It decreased from $VP_{med} = 0.1$ ($n = 10$) to $VP_{med} = 0.08$ ($n \geq 25$). The initial (lowest) class width *a*, which decreased with *k*, was less than 0.03. Similarly, the factor *b*, scaling the increase in interval-width

from one class to the next, decreased ($b = \{5.0, 3.4, 2.6\}$).

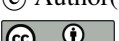



**Table 2.** Frequency intervals, derived from the distribution of *very poor* RB stability. The respective upper interval limits, as well as the most and second most frequent danger level the samples were drawn from (D($1^{st}$), D($2^{nd}$), respectively) are shown. All values for $n$ = 25. For $k$ = 4 classes, the terms used later in the manuscript are shown.

| classes $k$ | upper limit | D($1^{st}$) | D($2^{nd}$) | term |
|---|---|---|---|---|
| 3 | 0.031 | 1 | 2 | |
| | 0.159 | 2 | 3 | |
| | 1 | 4 | 3 | |
| 4 | 0.018 | 1 | 2 | *none or nearly none* |
| | 0.062 | 1 | 2 | *a few* |
| | 0.21 | 3 | 2 | *several* |
| | 1 | 4 | 3 | *many* |
| 5 | 0.013 | 1 | 2 | |
| | 0.034 | 1 | 2 | |
| | 0.089 | 2 | 1 | |
| | 0.237 | 3 | 2 | |
| | 1 | 4 | 3 | |

The thresholds of the class interval widths therefore depended primarily on $k$ rather than $n$. The resulting interval bin-widths for an exemplary value of $n$ = 50 and $k$ = {3, 4, 5} are shown in Table 2. In all cases, an additional class boundary would exist, generally at values between 0.5 and 0.9. As this class would remain empty most of the time, it is not shown in Table 2.

The correlation between the frequency class and the danger level was strong ($n$ = 50, $\rho$ > 0.83, p < 0.001). Even with $n$ = 10, with a large amount of overlap between classes, the correlation between frequency class and danger level was significant (RB: $\rho$ > 0.7, p < 0.001) The correlation increased with increasing $k$ and individual classes classified best for the respective lowest and highest frequency classes.

Using $k$ = 4 and the respective thresholds in Table 2, the median proportion *very poor* stability observed in each fre-
quency class were 0, 0.04, 0.12, 0.32.

## 4.2  Avalanche size

Most avalanches were of size 1 (Fig. 5a), except at 4-High, where a similar proportion of size 1, 2 and 3 avalanches were reported. The proportion of size 1 avalanches decreased with danger level, while the proportion of size 3 and 4
avalanches was highest at 4-High. Comparing the distributions at 1-Low to 3-Considerable shows that the most frequent avalanche size has little discriminating power to differentiate between danger levels. The median avalanche size was size



**Figure 4.** Comparison of observed (points, N = 41) and boot-strap sampled distributions (boxes) for the proportion of *very poor* (a, d), *very poor* and *poor* combined (b, e) and *good* stability tests (c, f), for two settings of the number *n* of tests drawn. When 7 to 15 RB tests were observed on the same day and within the same region, these are shown together with sampling using *n* = 10. When more than 16 tests were collected, these are shown together with *n* = 25. For *n* = 10 and *good* stability, the observed distributions were significantly different than the sampled distributions at 2-Moderate and 3-Considerable (p < 0.05, Wilcoxon rank sum test).





at 1-Low and 2-Moderate, and size 2 at 3-Considerable and 4-High (Fig. 5a).

The size distributions of the reported avalanches differed between the countries: size 1 were proportionally more frequent in SWI than in NOR (30% vs. 17%), while size 4 avalanches had larger proportions in NOR (NOR 2%, SWI 1%; see also Fig. E1a, b in the Appendix). Even though the proportion of reported size 1 avalanches decreased with increasing danger

level, size 1 was clearly the most frequently reported size at danger levels 1-Low to 3-Considerable in SWI, and at 1-Low in NOR (Fig. E1a, b). In Norway, size 2 avalanches were the most frequent size at 3-Considerable and at 4-High. At 3-Considerable in NOR, and at 4-High in both countries, about one third of the avalanches were size 3 or 4. In SWI, the distributions were almost identical for natural and for human-triggered avalanches. In NOR, there were proportionally more human-triggered size 1 avalanches than natural avalanches, for sizes 3 and 4 the opposite was the case.

Considering the size of the largest reported avalanche per day and warning region showed again rather similar size distributions at 1-Low and 2-Moderate (Fig. 5b). The median largest avalanche per day and region was size 2 for 1-Low to 3-Considerable, except at 4-High with size 3. However, the proportion of days when size 1 avalanches were the largest observed avalanche decreased considerably with increasing danger level, while the proportion of days with at least one size 3 or size 4 avalanche increased monotonically. At 4-High, more than 75% of the days had at least one avalanche of

size 3 or 4 recorded. This proportion was higher in SWI (78%; Fig. E1d) than in NOR (59%; Fig. E1c).

The correlation between the size of the avalanche and the danger level was weaker for the median size per day and warning region ($\rho = 0.15$, p $< 0.001$) than for the largest size ($\rho = 0.25$, p $< 0.001$).

The number of reported avalanches per day and warning region increased with danger level from 2.5, 4, 5 to 8 for 1-Low to 4-High, respectively. It is of note that we did not explore days with no avalanches as we were interested in the size of

avalanches, not their frequency. The frequency component is addressed using the frequency of locations with *very poor* stability as a proxy.

### 4.3   Combining snow stability, its frequency and avalanche size

Combining the snow stability class, its frequency and avalanche size considers all three key factors characterizing the avalanche danger level. We explored a data set consisting of the Swiss RB and avalanche data only:

– The number of frequency classes was set to $k = 4$ with $B = 2{,}500$ repetitions for each danger level. For this example, we selected the largest $n$ with a uni-modal histogram ($n = 25$).

– We classified the proportion of *very poor* stability using the thresholds and the four terms (*none or nearly none*, *a few*, *several* and *many*) for the 4 classes (Tab. 2).

– Each sample was complemented with an avalanche size, drawn from the distribution of the largest avalanche size

30       per day and warning region, for the respective danger level (Fig. 5b).

The resulting simulated data set contained the following information: *danger level, frequency class describing occurrence of very poor stability, largest avalanche size*. These data looked like the following, here for 1-Low:




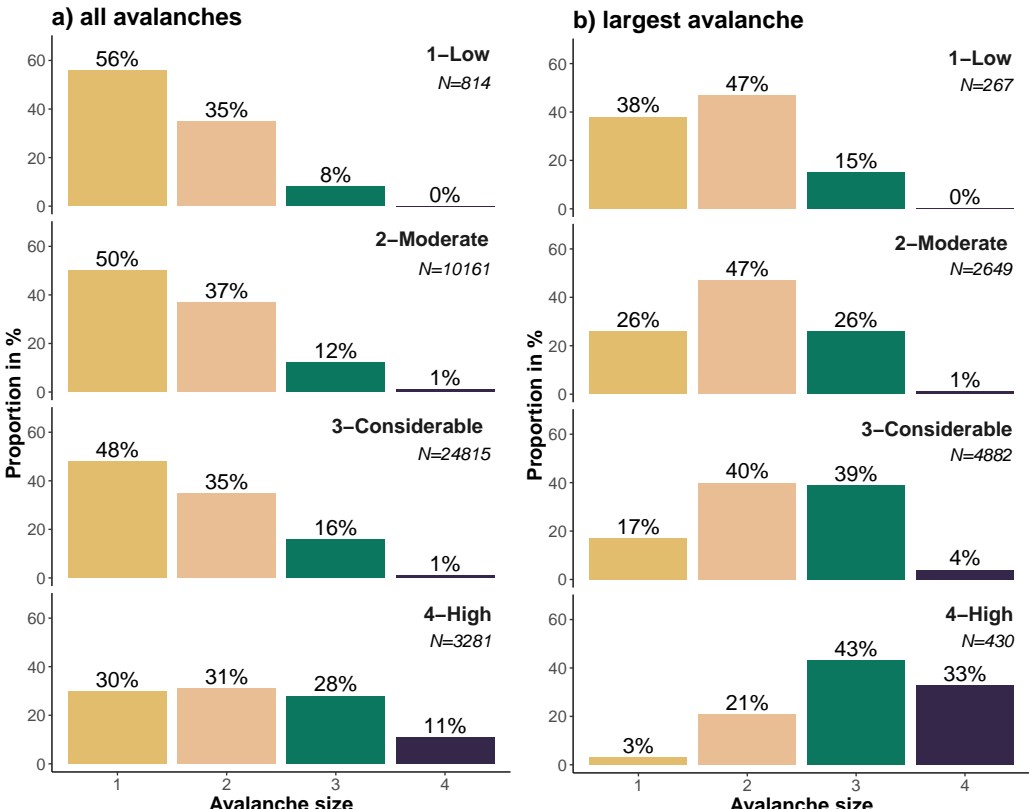

**Figure 5.** Size distribution of dry-snow avalanches, which released naturally or were human-triggered for danger levels 1-Low to 4-High, showing (a) all avalanches and (b) the largest reported avalanche per day and warning region. Note, proportions reflect the combined number of avalanches (more avalanches in SWI than in NOR).

*Sample 1: 1-Low, a few, avalanche size 1*

*Sample 2: 1-Low, none or nearly none, avalanche size 2*

*Sample 3: 1-Low, a few, avalanche size 1*

. . .

5 *Sample B: 1-Low - none or nearly none - avalanche size 1*

Tab. 3 summarizes the simulated data set. The most frequent combinations of the frequency class and avalanche size for each danger level were:

- 1-Low: *None or nearly none* locations with *very poor* stability (53%) exist. The largest avalanches are size 2 (48%).

10 - 2-Moderate: *A few* locations with *very poor* stability (37%) are present. However, *none or nearly none* or *several* locations are of almost similar frequency (32-31%). The typical largest avalanche is of size 2 (50%).





**Table 3.** Table showing the combination of the frequency class of *very poor* snow stability and the largest avalanche size for the four danger levels. Frequencies are rounded to the full per cent value. "–" indicates that these combinations did not exist.

| size | 1-Low | | | | 2-Moderate | | | | 3-Considerable | | | | 4-High | | | |
|---|---|---|---|---|---|---|---|---|---|---|---|---|---|---|---|---|
| | none* | few | several | many | none* | few | several | many | none* | few | several | many | none* | few | several | many |
| 1 | 17 | 10 | 5 | – | 8 | 9 | 7 | 0 | 0 | 2 | 12 | 2 | – | 0 | 0 | 1 |
| 2 | 25 | 16 | 7 | – | 16 | 19 | 15 | 0 | 1 | 3 | 30 | 5 | – | 0 | 3 | 18 |
| 3 | 11 | 8 | 3 | – | 8 | 9 | 9 | 0 | 1 | 3 | 30 | 6 | – | 0 | 6 | 37 |
| 4 | – | – | – | – | 0 | 0 | 0 | 0 | 0 | 0 | 3 | 1 | – | 0 | 5 | 30 |

\* - *none or nearly none*

simulation setting: Rutschblock, avalanches (SWI), *n* = 25, *k* = 4, *B* = 2,500 per danger level

- 3-Considerable: *Several* locations with *very poor* stability (75%) exist. The typical largest avalanches are sizes 2 or 3 (79%).

- 4-High: *Many* locations with *very poor* stability (86%) exist. The typical largest avalanche is of size 3 (43%).

Tab. 4 summarizes the simulated stability class - frequency class combinations for all stability classes, and the respective
5  most frequent and second most frequent danger level. The frequency class describing *very poor* stability was closely linked to one or two danger levels, which reflects Tab. 3. *Poor* stability as the most unstable stability class (when *none or nearly none very poor* existed), was generally associated with 2-Moderate or 1-Low. If both *very poor* and *poor* stability fell into the category *none or nearly none*, the resulting danger level was mostly 1-Low. The actual danger level distributions, summarized in Tab. 4, are shown in the Appendix (Fig. G1).

Finally, we present a data-driven look-up table to assess avalanche danger (Fig. 6) using the simulations presented before. We used a step-wise approach, and two matrices as proposed by Müller et al. (2016) in the so-called Avalanche Danger Assessment Matrix (ADAM). In a first step, the most unfavorable snowpack stability class is combined with its frequency (Fig. 6, left matrix, which we refer to as *stability matrix*). The resulting most unfavorable stability class -
15  frequency class combination, which has a frequency greater than *none or nearly none* (>1.8%, Tab. 2), is retained. Cell labels (letters A to E) shown in the *stability matrix* correspond to similar danger level distributions related to this most unfavorable stability class - frequency class combination according to Table 4. The simulated RB stability class distributions behind the cells A-E are shown in Figure 7. In a second step, the most appropriate cell describing stability and its frequency (letter in the *stability matrix*) is combined with avalanche size (Fig. 6, right matrix, which we refer to as
20  *danger matrix*). The *danger matrix* displays the most frequent danger level (bold) and the second most frequent danger level (if present more than 30%: no brackets, if present between 15 and 30%: in brackets) characterizing this combination. Again, to illustrate the actual danger level distributions behind this matrix, Figure 8 summarizes the simulated data.





***stability matrix***

| snowpack stability | frequency: none* | few | several | many |
|---|---|---|---|---|
| very poor | ** | D | B | A |
| poor | ** | E | D | C |
| fair | - | - | E | E |
| good | - | - | - | - |

* none or nearly none
** if none, refer to next higher stability class
\- no data
C cell contains less than 1% of the data

***danger matrix***

| stability matrix | largest avalanche size: 1 | 2 | 3 | 4 |
|---|---|---|---|---|
| A | 3 -4 | 4 (-3) | 4 | 4 |
| B | 3 (-2/-1) | 3 (-2) | 3 (-2) | 4 -3 |
| C | 2 (-3) | 2 -3 | 3 -2 | - |
| D | 1 -2 | 2 -1 | 2 -1 | 3 (-2) |
| E | 1 | 1 (-2) | 1 (-2) | - |

-3: >30%
(-3): 15-30%

**Figure 6.** Data-driven look-up table for avalanche danger assessment (similar to the structure proposed by Müller et al. (2016)). Refer to text for details.

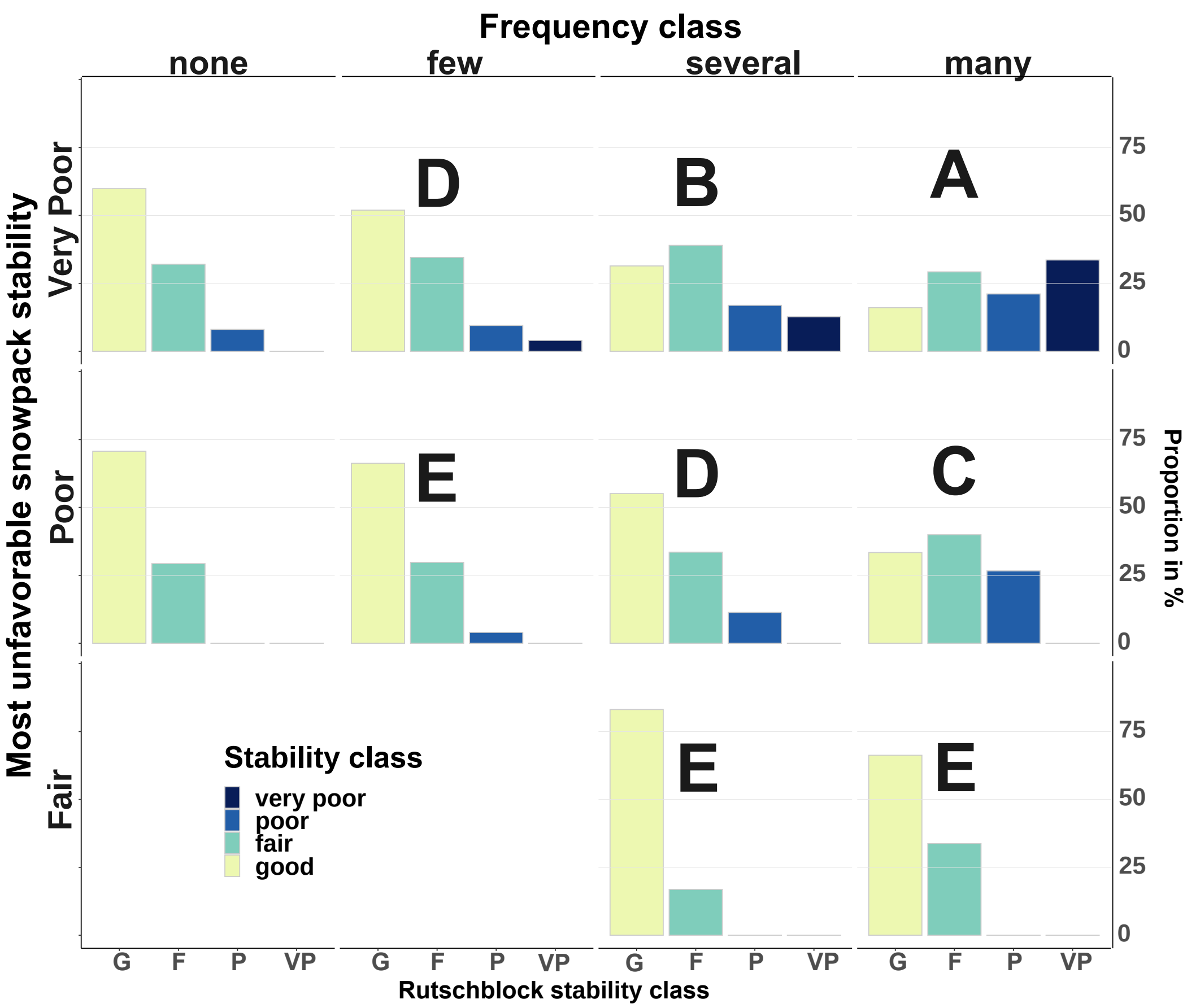

**Figure 7.** Mean simulated RB stability classes behind the *stability matrix* in Figure 6. Letters describe cells with the corresponding most frequent and second most frequent danger level (Tab. 4 and Fig. G1).





**Table 4.** Summary of the simulated RB stability and frequency class combinations, and the respective most frequent danger level D(1st) and the second most frequent danger level D(2nd). Combinations of stability and frequency classes resulting in the same D(1st) and D(2nd) are indicated by the same letters in the *group*. Letters are ordered according to rank-order of D(1st) and D(2nd). If a frequency class is *none or nearly none*, the next higher stability class should be considered. The data behind this summary table is shown in Fig. G1 in the Appendix.

| stability | frequency class | D(1st) | D(2nd) | group |
|---|---|---|---|---|
| *very poor* | *many* | 4 | 3 | A |
| | *several* | 3 | 2 | B |
| | *few* | 2 | 1 | D |
| | *none\** | check *poor* stability | | |
| *poor* | *many* | 2 | 3 | C |
| | *several* | 2 | 1 | D |
| | *few* | 1 | 2 | E |
| | *none\** | check *fair* stability | | |
| *fair* | *many* | 1 | 2 | E |
| | *several* | 1 | – | F |

*\* - none or nearly none*
All values for RB, $n = 25$, $k = 4$, $B = 2,500$ per danger level

## 5 Discussion

In the following, we discuss our findings in the light of potential uncertainties linked to the data (Sect. 5.1) and methods selected (Sect. 5.2). Furthermore, we compare the results to currently used definitions, guidelines and decision aids used in regional avalanche forecasting (Sect. 5.3).

### 5.1 Data

We relied on observational data recorded in the context of operational avalanche forecasting. This means that differences in the quality of single observations are possible. For instance, variations in both the estimation of avalanche size (Moner et al., 2013) as well as in locally assessing the avalanche danger level (Techel and Schweizer, 2017) have been noted. Furthermore, observations of avalanche activity often have a temporal uncertainty of a day or more, especially in situations with prolonged storms and poor visibility that often accompany a higher danger level. We addressed these issues by filtering the most extreme 2.5% of the avalanche observations for each danger level.

Completeness of observations is another issue. Avalanche recordings are generally incomplete, in the sense that not all avalanches within an area are recorded as well as that single observations may lack information, e.g. on size. However, the size distributions (Fig. 5) reflect that smaller avalanches are more frequent, which was also observed in previous



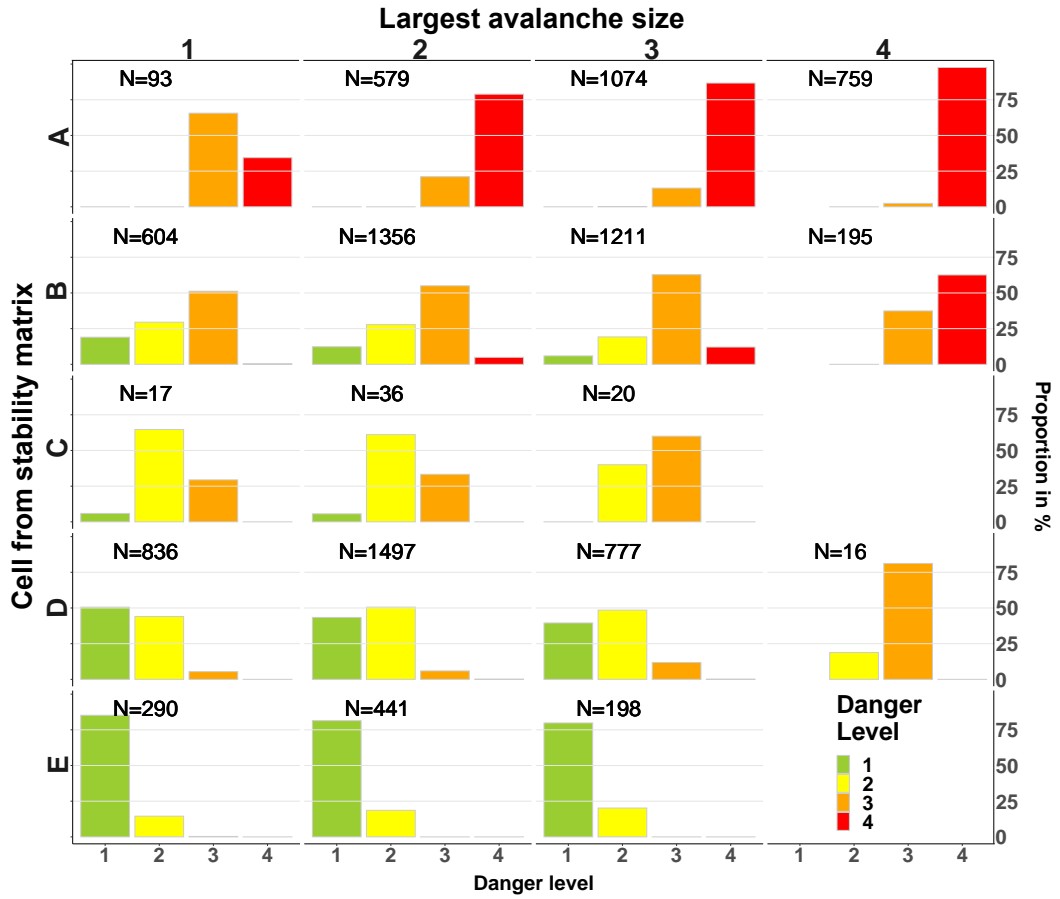

**Figure 8.** Distribution of danger levels for combinations of the typical largest avalanche size and the cells obtained before in the *stability matrix* (A-E, Fig. 6 left). The most frequent and second most frequent danger levels in each cell - avalanche size combination are shown in the *danger matrix* in the right part of Fig. 6.

studies where other recording systems were applied such as recording of avalanches by snow safety staff and the public (Logan and Greene, 2018), manual mapping of avalanches (Hendrikx et al., 2005; Schweizer et al., 2020) or satellite-detection of avalanches (Eckerstorfer et al., 2017; Bühler et al., 2019). Still, smaller avalanches may be underrepresented compared to larger avalanches - as was the case for instance for size 1 avalanches in NOR (Fig. E1a). This underre-

5 porting may depend on the relevance to an observer, but also on the ease of recording or limitations set by the recording of numerous smaller avalanches. As we did not primarily use the number of avalanches, but focused on the largest avalanche per day and warning region, we expect this limitation to be less relevant.

Stability tests conducted by specifically trained observers are often performed at locations, where the snowpack is expected to be weak, though in an environment where spatial variability of the mountain snowpack can be high (e.g.

10 Schweizer et al., 2008a). Additionally, in most cases just one stability test was performed by an observer, not permitting





us to judge whether this test was representative for the conditions of the day. However, the overall distributions of the stability test results, regardless whether RB or ECT were considered (Fig. 3), highlight the increase of locations with low snow stability at higher danger levels.

To address potential bias in observations linked to a specific warning service (e.g. Techel et al., 2018), we used data from

two different warning services (NOR, SWI). This brought additional challenges, like a different structure or content of the observational data, which required us to make further assumption (e.g. for counting the number of avalanches reported in forms when several avalanches were reported together in Norway). The stability distributions of the ECT (Fig. A1) or the largest avalanche size per day and warning region (Fig. E1) showed similar overall patterns across countries, with increasing frequencies of *very poor* stability and increasing avalanche size with increasing danger level.

At 4-High, stability test data were limited, as these situations are not only rare and temporally often short-lived, but also since backcountry travel in avalanche terrain is dangerous and therefore not recommended. As a consequence, not only considerably fewer field observations were made, but these were also dug on less steep slopes at lower elevation, which may potentially underestimate snow instability.

Finally, stability test results, avalanche observations and local danger level assessment are generally not independent

from each other, as often the same observer provided all this information. However, as shown by Bakermans et al. (2010), stability test results – compared to other observations - have relatively little influence on a local danger level estimate, while numerous natural or artificially triggered avalanches are a clear indication for a higher danger level and may thus raise the quality of the local assessment.

## 5.2    Methods

### 5.2.1    Stability classification of RB and ECT

We relied on existing RB and ECT classifications (RB: Schweizer and Wiesinger (2001); Schweizer (2007a); ECT: Techel. et al. (in prep.), Fig. 1). While the RB classification scheme is well-established in the operational assessment of snow profiles in the Swiss avalanche warning service, the classification of ECT into four stability classes has only recently

been proposed by Techel. et al. (in prep.). They showed that for a large data set of pairs of ECT and RB performed in the same snow pit, both classifications provided good correlations to slope stability. However, the most favorable and the most unfavorable RB stability classes captured slope stability better than the respective ECT classes, indicating a lower agreement between slope stability and ECT results compared to the RB. This was our argument for not fully aligning the four RB and ECT stability classes and is supported by our findings: The RB stability class distributions changed more

pronounced from 1-Low (68% *good* stability, 2% *very poor*) to 4-High (10% *good*, 38% *very poor*) than the most favorable and unfavorable ECT stability classes (1-Low: 60% *good* stability, 8% *poor*, 4-High: 15% *good*, 23% *poor*).




### 5.2.2  Simulation of stability distributions

We could not rely on a large number of stability distributions observed on the same day in the same region, which is a general problem in avalanche forecasting. We therefore generated stability distributions using re-sampling methods (Sect. 3.2) and by selecting sampling settings which lead to considerably overlapping distributions (Fig. 4). We argue that

some overlap in stability distributions would characterize the large variability of avalanche conditions. However, which $n$ captures the variation best, we do not know. We suspect that a combination of (labour-intensive) field measurements combined with spatial modeling in a large variety of avalanche conditions will be necessary to shed some light on this question (e.g. Reuter et al., 2016, for a small basin in Switzerland). Alternatively, spatial modeling of the snowpack, provided that a robust stability parameter can be simulated, would be required.

Repeated sampling from small data sets may underestimate the uncertainty associated with a metric, but more importantly, the question must be raised, whether the sample reflects the population well. While at 1-Low to 3-Considerable, we sampled from between 700 and 2800 RB or ECT tests per danger level, at 4-High the respective number of observations was very small (RB: N = 21, ECT: N = 13). Hence, both the data shown in Fig. 3 as well as the sampled stability distributions for this danger level are more uncertain than for the other danger levels. While the combined number of

locations with *very poor* and *poor* stability increased, and those with *good* stability decreased at 4-High (Fig. 3), judging whether the observed tests reflect the population well is difficult. For instance, when exploring the very small ECT data sets for the two countries individually (NOR: N = 6, SWI: N = 7; Fig. A1), the uncertainties associated with very small data sets are highlighted. Unfortunately, we are not aware of other studies, which have explored the snow stability distribution in a region at 4-High based on many tests, and therefore have no comparison. Even on 7 Feb 2003, one of the days

of the verification campaign in the region of Davos/SWI (Schweizer et al., 2003), the forecast danger level 4-High was only «verified» to be between 3-Considerable and 4-High (Schweizer, 2007b). On this day, 14 Rutschblock tests were observed. 36% of these were either *very poor* or *poor*, thus being close to the average values noted for 3-Considerable (Fig. 3a). It is of note that these data was not considered in our analysis, as we analyzed only stability data when only one specific danger level was locally estimated.

Comparing the distributions of our snow stability classes with the characteristic stability distributions obtained during the verification campaign in Switzerland in 2002 and 2003, some differences can be noted (Swiss RB data): For instance, the proportion of *very poor* and *poor* combined was at 2-Moderate about 15% and at 3-Considerable about 40%, which is lower than Schweizer et al. (2003)'s findings (20-25% and about 50%, respectively). At 1-Low, about 70% of the RB tests were classified as *good*, while Schweizer et al. (2003) noted about 90% of the profiles to have *good* or *very good*

stability. This suggests a smaller spread in the distribution of our automatically assigned stability classes, compared to the manual classification approach according to Schweizer and Wiesinger (2001).





### 5.2.3 Classification of snow stability frequency distributions

In addition to simulating snow stability distributions using a re-sampling approach, we attempted for the first time a data-driven classification of the proportion of *very poor* stability tests. Our approach shows that the number $n$ drawn for each bootstrap has little influence on class interval definitions, as long as the resolution of the test statistic is sufficiently high.

Class thresholds are primarily defined by the central tendency of the distribution, in our case the median proportion of *very poor* stability tests $VP_{med}$, and by the number of classes preferred $k$. In the case of a low resolution of the test statistic the class interval widths should be scaled according to the number of distinct measurements (Evans, 1977). In other words, with $n = 10$ and $b = 2$, class interval widths would be [0], [0.1, 0.2], [0.3, 0.4, 0.5, 0.6], . . .

Assigning a class to the proportion of *very poor* stability, however, was affected by $n$ due to the fact that $n$ influences both
the resolution of the statistic and the variance, while the overall class assignment was less dependent on $n$. This means that conceptually we can think in frequency classes, as long as class interval boundaries are scaled according to the data used. Furthermore, the simulated stability distributions indicate that the focus is on optimizing class definitions to values between 0 and 40% when relying on stability tests, rather than the entire potential parameter space (0-100%).

The preferred number of classes $k$ cannot be defined and may depend on a number of factors. We suggest that defining
$k$ should be guided by keeping classes as distinguishable as possible - for instance by addressing the frequently occurring low proportions of *very poor* stability on one side and the rarely observed large proportions of *very poor* stability on the other side, and potentially a class covering the in-between. Furthermore, these terms must be unambiguously understandable to the user, regardless of language.

### 5.3 Data interpretation

### 5.3.1 Snowpack stability and its frequency

We showed an increasing number of locations with *very poor* snow stability with increasing danger level, which is in line with previous studies exploring point snow stability within a region or small basin (Schweizer et al., 2003; Reuter et al., 2016) or the number of natural and human-triggered avalanches within a region (e.g Schweizer et al., 2020). This
correlation was generally strong, and even when using a sampling setting leading to large variation and overlap ($n = 10$) and a small number of classes $k$, the correlation between the frequency class describing *very poor* stability and the danger level was still moderate (Sect. 4.1.2).

We explored snowpack stability using RB and ECT, which describe the stability at a specific point. However, within a slope or a region, point snow stability is variable (e.g. Birkeland, 2001; Schweizer et al., 2008a). This can be expressed by the
frequency a certain stability class exists, and by additionally describing the locations more specifically. When describing the avalanche danger level in a region, snowpack stability and its frequency distribution are therefore inseparable. We suggest that primarily the frequency of the lowest stability class is relevant for the assignment of a danger level, as this stability class combined with its frequency describes the minimal trigger needed to release an avalanche and how frequent




these most unstable locations exist within a region. The combination of stability class and its frequency distribution will also define which aspects and elevations should be described with the same danger level. Furthermore, the specific description of triggering locations, for instance *at treeline* or *in extremely steep terrain*, may provide an indication where in the terrain these locations may exist more frequently within its frequency class. Even though different terms are used,

both the EAWS-Matrix (**?**) and the CMAH (Statham et al., 2018a) first combine snowpack stability and its frequency distribution, before avalanche size is considered. The respective terms which were used are the 'load' (trigger) and the 'distribution of hazardous sites' in the EAWS-Matrix and the 'sensitivity to triggers' and 'spatial distribution' leading to the 'likelihood of avalanches' in the CMAH.

We explored primarily the frequency of the stability class *very poor*, which is most related to actual triggering points.

However, as several studies have shown, even when stability tests suggested instability, often only some of the slopes were in fact unstable and released as an avalanche (e.g. Moner et al., 2008; Techel. et al., in prep.). Thus, depending on the data used to define *very poor* stability, for instance whether stability tests are used or natural avalanches, an adjustment of class intervals may be necessary for it to capture the frequency of locations where natural avalanches may initiate or where human-triggered avalanches are possible.

### 5.3.2   Avalanche size

The most frequent avalanche size had little discriminating power, with the typical size being of size 1 or size 2, regardless of danger level. This finding is similar to other studies (Harvey, 2002; Logan and Greene, 2018; Schweizer et al., 2020). All three studies showed that the typical avalanche size did not increase with danger level, except at 4-High in the study by Logan and Greene (2018).

We showed that considering the largest avalanche per day resulted in a slightly better discrimination between danger levels. This finding is also supported by Schweizer et al. (2020), with the size of the largest avalanche being mostly of size 4 at 4-High. Furthermore, the typical largest expected avalanche is highly relevant for risk assessment and mitigation. For danger level 5-Very High, for which we had no data, other studies have shown a further shift towards size 4 avalanches. Schweizer et al. (2020) showed that at 5-Very High size 4 avalanches were 15 times more frequent than

at 3-Considerable and five times more frequent compared to 4-High. In two extraordinary avalanche situations in January 2018 and January 2019, when danger level 5-Very High was verified for parts of the Swiss Alps, avalanches recorded using satellite data showed that often ten or more size 4 avalanches and/or one size 5 avalanche was observed per 100 km$^2$ (Bühler et al., 2019; Zweifel et al., 2019).

### 5.3.3   Combining snow stability, its frequency and avalanche size

In Section 4.3 we presented a data-driven look-up table to assess avalanche danger (Fig. 6). As can be seen in this table, the combination of snow stability and its frequency that best matches an avalanche situation (A to E), is highly relevant for danger level assessment. In general, avalanche size only has a rather minor influence on the danger level,





once the cell describing stability has been fixed. This is in contrast to the original avalanche danger level assessment matrix (ADAM, Müller et al., 2016) that proposed that an increase in either the frequency class or the avalanche size, or a decrease in snow stability, should lead to an increase in danger level by one level. Clearly, the presented data-driven look-up table highlights that a greater focus must be placed on snow stability and its frequency distribution, compared

to avalanche size, when assessing avalanche hazard. This was also shown by Clark and Haegeli (2018), who explored the combination of descriptive terms describing the three factors in the data behind the avalanche forecasts in Canada and their relation to the published danger level and avalanche problem. They showed that the 'likelihood of avalanches', which compares to our *stability matrix* (Fig. 6), also had a greater impact on the resulting danger level than avalanche size, even though avalanche size ≤1.5 (considered harmless to people) was often a first split in a decision tree model.

Hence, despite using different approaches, partially different terminology and slightly different avalanche danger scales in Europe and North America, the relative importance of the three key contributing factors and the distributions of the danger levels are similar.

Our approach can only provide general distributions observed under dry-snow conditions. The look-up table presented here should therefore primarily be seen as a tool stimulating not only the discussion of specific situations but also when

attempting to improve the definitions underlying the categorical descriptions of the danger levels.

## 6 Conclusions

We explored observational data from two different countries relating to the three key factors describing avalanche hazard, snowpack stability, its frequency distribution and avalanche size. We simulated stability distributions and defined classes summarizing the frequency of potential avalanche triggering locations.

Our findings suggest that the three key factors did not distinguish equally prominently between the danger levels:

- The proportion of *very poor* or *poor* stability test results increased from one danger level to the next higher (Figures 3 and 4). Considering *very poor* snowpack stability and its frequency alone, already distinguished rather well between danger levels 2-Moderate, 3-Considerable and 4-High (Tab. 4).

- Considering the largest observed avalanche size per day and warning region was most relevant to distinguish
between 3-Considerable and 4-High (Fig. 5 and Tab. 3). For other situations, the largest avalanche size - used by itself - had comparably little discriminating power at 1-Low to 3-Considerable (Fig. 5).

In summary, the frequency of the most unfavorable snowpack stability class is the dominating discriminator. At higher danger levels the occurrence of size 4 avalanches discriminates danger level 3-Considerable from 4-High. We further suppose that the occurrence of size 5 avalanches discriminates between 4-High and 5-Very High without an additional

increase in the spatial distribution of *very poor* stability. This shift in importance between factors is currently poorly represented in existing decision aids like the EAWS-Matrix or ADAM (Müller et al., 2016), but also in the European Avalanche Danger Scale.





We hope that our data-driven perspective on avalanche hazard will allow a review of key definitions in avalanche forecasting, like the avalanche danger scale.

*Data availability.* The data will become freely available at www.envidat.org.

*Author contributions.* FT designed the study, conducted the analysis, wrote the manuscript. KM extracted the Norwegian data. KM and
5   JS repeatedly provided in-depth feedback on the study design and analysis, and critically reviewed the entire manuscript several times.

*Competing interests.* No competing interests.

*Acknowledgements.*



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





## 1 Appendix: ECT - simulated snow stability distributions and frequency classification

As a supplement to the analysis shown for the RB in the main part of the paper, in the following we show the key results for the ECT. As for the RB, we tested $n$ = {10, 25, 50, 100, 200, 1000}. Besides visual inspection, we additionally tested the *poor* stability distributions for multi-modality using the *modetest* (Ameijeiras-Alonso et al., 2018).

In contrast to the RB class *very poor* stability, the distribution of the proportion of *poor* ECT stability was less skewed towards lower proportions of *poor* stability. Increasing $n$ impacted the number of modes detected in the histograms, with two or more modes being present when $n$ reached values of about 100 (Fig. B1g-l). Exploring the bootstrapped-sampled distributions for the most extreme ECT stability classes *poor* and *good* (Fig. D1) generally showed similar results as for the RB (Fig. C1). However, while the distributions for the RB also exhibited a logical pattern at 4-High (Fig. C1f), despite

being drawn from a small population (drawn from N = 21), the same cannot be noted for the ECT (Fig. D1f, drawn from N = 13).

Comparing the sampled distributions with actually observed distributions of stability tests on the same day and in the same region (N = 31), showed that the distributions obtained using bootstrap-sampling reflected the variation in the observed distributions not always well (Fig. F1). Visually comparing the results for $n$ = 10, where there was still a reasonably

large number of days with 7 to 15 ECT (N = 5, 6, 9), implies that the bootstrap-sampled distributions captured the observed distributions poorly. However, a significant deviation between sampled and observed distributions was only noted for *good* stability at 3-Considerably (p = 0.02, Fig. F1c). It must be noted, however, that sample sizes are small impacting both the likelihood to obtain unusual data sets in the field as well as for p-values not being the optimal indicator to detect significant differences.





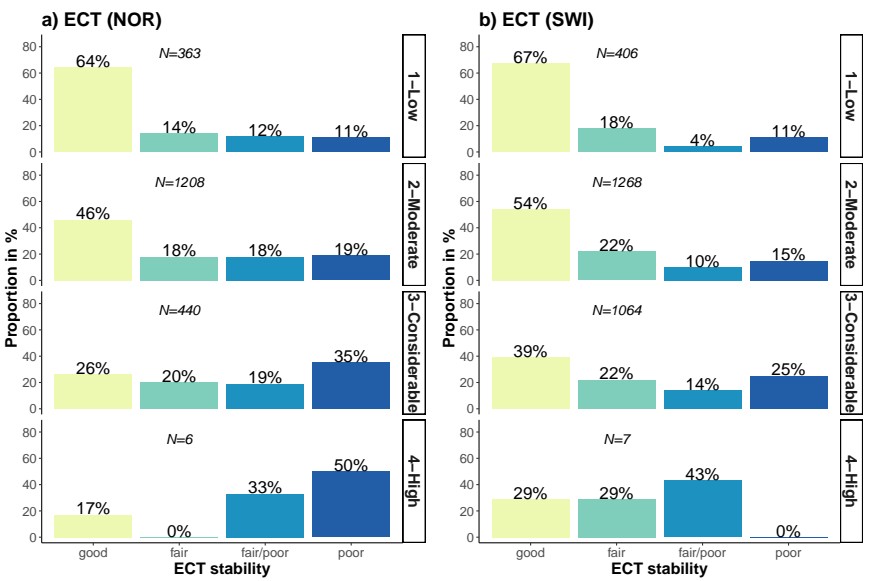

**Figure A1.** Bar plots showing distribution of stability ratings for ECT for (a) Norway and (b) Switzerland. Note the very small number of tests at 4-High. The ECT classification scheme is shown in Fig. 1b.

## 2 Appendix: Additional figures and tables





**Figure B1.** Simulated proportions of *very poor* (RB) or *poor* (ECT) stability for different number of samples *n* drawn in each of the bootstraps for (a-f) Rutschblock and (g-l) ECT. The more samples drawn, the more the data becomes multi-modal and clustered around the means of each danger level. This is indicated by the p-value (*modetest*, median p-value of 10 repetitions, Ameijeiras-Alonso et al., 2018). See also Fig.s C1 and D1 for two-dimensional plots.



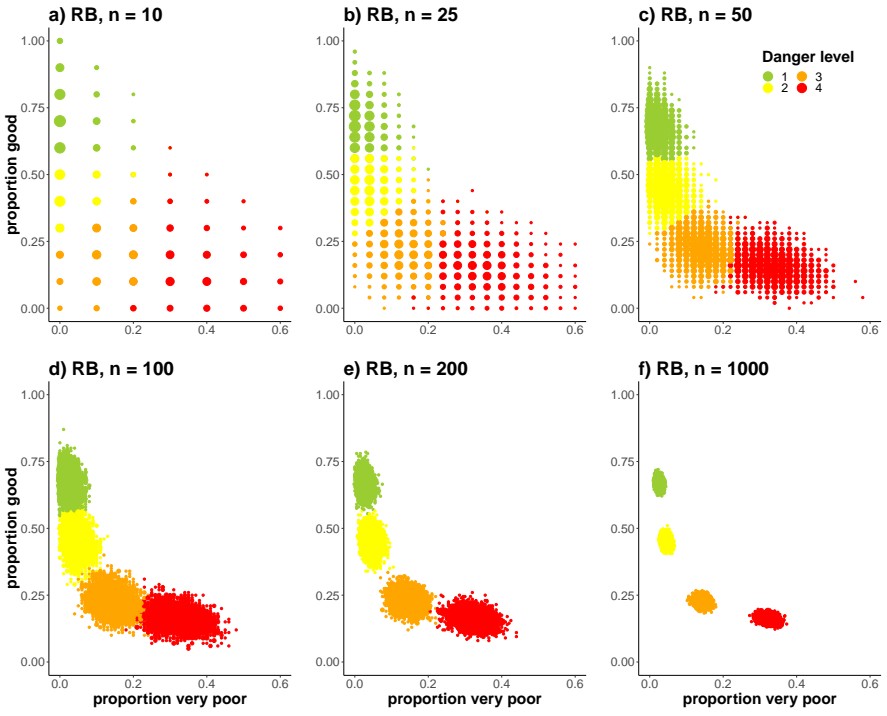

**Figure C1.** Simulated proportions of *very poor* (x-axis) and *good* RB-stability (y-axis), for different number of samples *n* drawn in each of the bootstraps (a-f). The colour represents the most frequent danger level for the respective *very poor - good* combination. The more samples are drawn, the more the data becomes clustered around the means of each danger level.



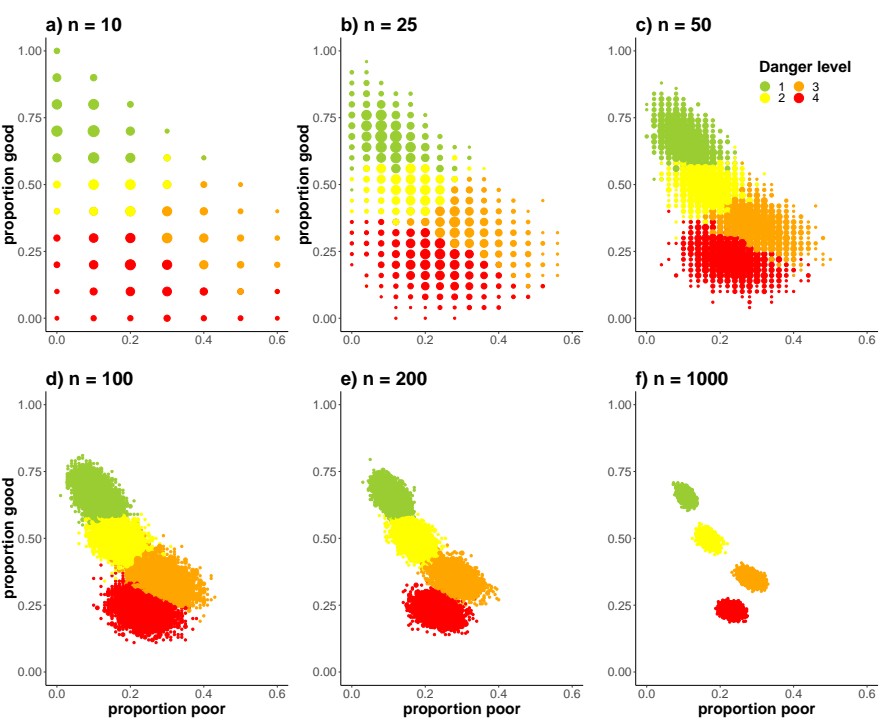

**Figure D1.** Simulated proportions of *poor* (x-axis) and *good* ECT-stability (y-axis), for different number of samples *n* drawn in each of the bootstraps (a-f). The colour represents the most frequent danger level for the respective *poor - good* combination. The more samples are drawn, the more the data becomes clustered around the means of each danger level.





**Figure E1.** Bar plots showing the size distribution of all avalanches (upper row) and the largest avalanche *per day and warning region (lower row), for Norway (left column) and Switzerland (right column).





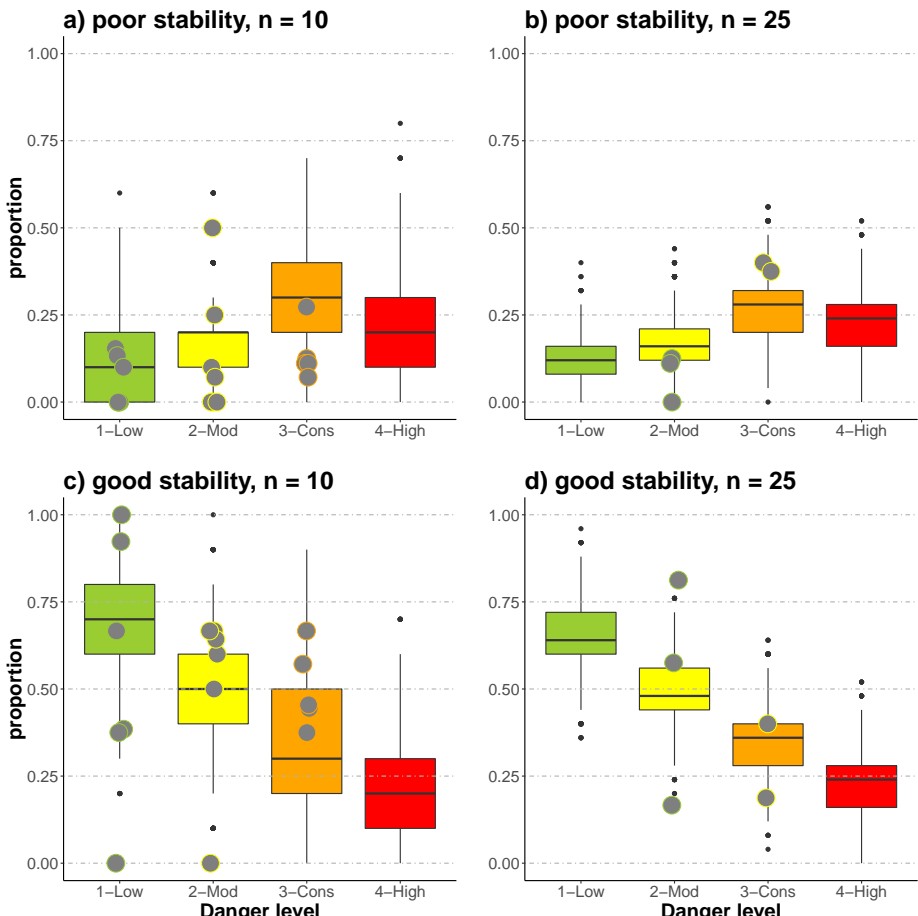

**Figure F1.** Comparison of observed (points, N = 31) and bootstrap-sampled ECT distributions (boxes) for the proportion of *poor* (a, b) and *good* stability tests (c, d), for two settings of the number *n* of tests drawn. Observations with 7 to 15 individual tests on the same day and within the same region are shown together with sampling using *n* = 10. When more than 16 tests were collected, these are shown together with *n* = 25.

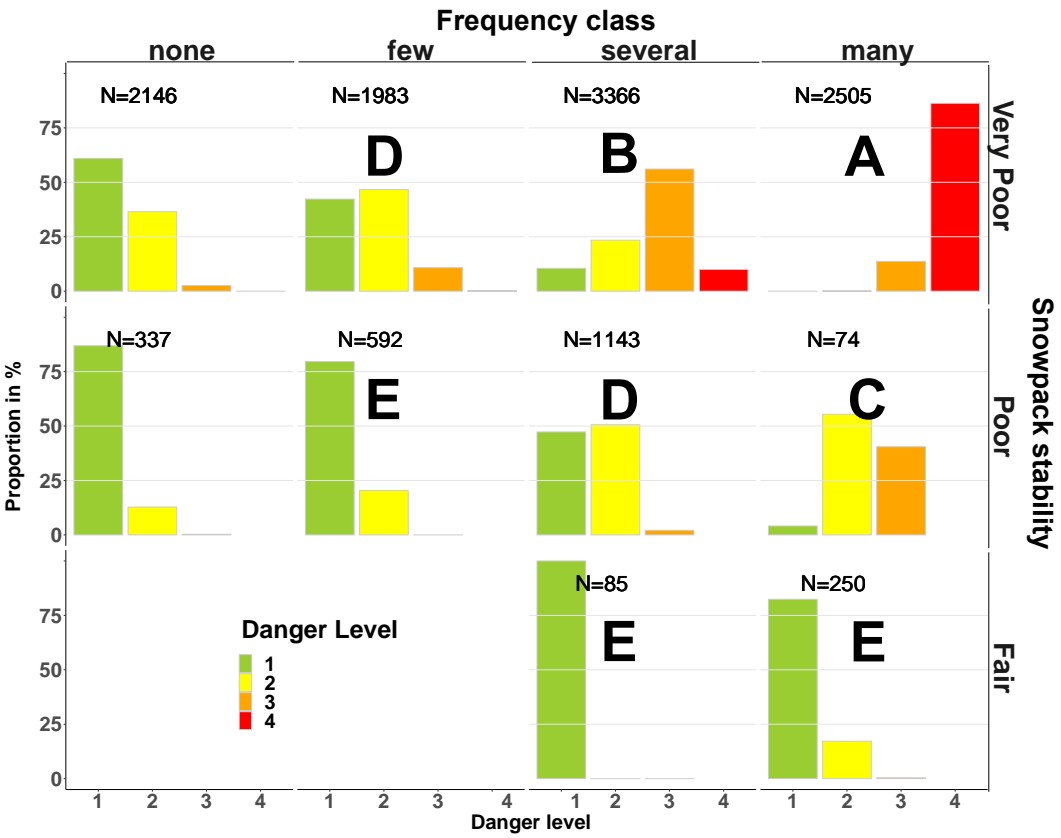

**Figure G1.** Distribution of danger levels for snowpack stability and frequency class combinations. Combinations with the same most frequent and second most frequent danger level are labelled with the same letter (A to E). If a lower stability class resulted in frequency class *none*, for these cases the distributions for the next higher stability class is shown in the respective row below (i.e. the 2146 cases of *none very poor* are shown in the row *poor*). The letter which comes first in the alphabet is retained and used as a reference for the following matrix (Fig. 8). This matrix corresponds to the stability matrix in Fig. 6.