# Peer review of "On the importance of snowpack stability, the frequency distribution of snowpack stability, and avalanche size in assessing the avalanche danger level"

_The Cryosphere, 2020_

## Referee Comment (RC1) · Simon Horton (Referee) · 22 Mar 2020

**General comments**

The paper addresses the subjective nature of avalanche hazard ratings by exploring a large dataset of hazard ratings and field observations. Since hazard ratings are defined by subjective terms, this study presents a unique approach to quantifying some of these terms. This could improve the consistency of hazard assessments and risk

communication, which is an important contribution. The scientific methods applied in the paper are rigorous, and the conclusions drawn from the results are appropriate and interesting.

My main comment is the key results of the study could be more clearly communicated and emphasized. I found there were some additional data and results that distracted from the key messages, and as a result I needed to read certain sections twice to properly understand the relevance. I think this could be easily addressed by restructuring and shortening some sections. My other main comment is I think more discussion about the application of the results (e.g. improved forecasting methods, danger rating definitions) would make the contributions clearer to the reader.

**Specific comments**

- Relevance of additional data sets: The key conclusions of the study appear to come from the Swiss Rutschblock test and avalanche data, however while reading the results there are numerous references to patterns between the Swiss/Norwegian and RB/ECT data. I found this distracting from the main research question about the contributing factors to danger ratings. I would consider restructuring some of the sections so the main research question is addressed first, and then perhaps distinct sub-sections discussing how the core results differ between SWI/NOR and between RB/ECT results. There are also quite a few Appendix figures with these additional data sets which disrupts the flow while reading the results.

- Applications/next steps: I found the discussion focused too much on the limitations of the study rather than focusing on an in-depth discussion of how the results relate to the main research question. I would suggest shortening the limitations and adding more discussion about how these results could be applied

to improve danger ratings. Linkages to existing hazard assessment frameworks (e.g. ADAM, CMAH) are discussed, but I would have found it interesting to read more about how the results could improve or unify the existing frameworks. For example, the stability matrix in Fig. 6 has many parallels with the likelihood matrix from the CMAH (Statham et al. 2018, Fig. 2), the Bavarian matrix, and ADAM. While these are discussed, I think the strong quantitative data in this study are well positioned to make an informed critique on existing methods and more suggestions for future directions.

- Applying stability distributions: While the bootstrap sampling method is appropriate to derive stability distributions and define classes, it is somewhat theoretical and not clear how the derived classes could be applied in practice. Can these distributions help us understand more about stability conditions for a given hazard level? For example, from Fig. 3 could we assume that roughly 17% of slopes are unstable at 3-Considerable and 38% of slopes are unstable at stable at 4-High? I understand the challenges with making that inference, however, I wonder if the numeric nature of the study could help give some additional meaning to terms like few, several, and many. And although the theoretical meaning of these terms are clearly defined, how can forecasters actually assess whether the frequency of unstable locations is few/several/many?

- Methods: The beginning of each sub-section could use a bit more context about how that step is relevant to exploring the link between danger ratings and a contributing factor.

- Figures: The figures are clear, legible, and support the main messages of the study well.

**Technical comments**

- p1 line 7: Although less precise, saying "frequency of unstable locations" may be simpler to understand when reading the abstract only

- p1 line 13-14: Consider adding "simulated stability distributions" (the snowpack distribution isn't simulated)

- p2 line7: Preferable to use consistent terminology from the list of key factors, i.e. "probability of avalanche release" instead of "release (or triggering) probability"

- p2 line 13: Similar to above, starting the paragraph by repeating the term "frequency and location of triggering spots" would make it clearer the paragraph ties back to the list of key factors

- p2 line 23: missing citation

- p2 line 24: According to the CMAH spatial distribution also considers spatial density. Statham et al. 2018: "Spatial distribution considers the spatial density and distribution of an avalanche problem and the ease of finding evidence to support or refute its presence."

- Table 1: The "data from" column heading isn't clear if the data is from just a single season or all seasons up to 2018/19 (as explained in footnote). Consider a more precise heading or list season ranges in the table (e.g. 2002-2019)

- p4 line 2-4: These two sentences aren't necessary, as they are discussed below.

- p6 line 4-5: Please be consistent with order of reporting SWI and NOR data, in this sentence NOR is described first.

[Figure]

- p6 line 8: It would be helpful to start this section by explicitly explaining the purpose of this step is to relate the snowpack test data to one of the explanatory factors in the study (i.e. probability of avalanche release)

- p6 lines 19-26: This is an example of how the addition of ECT data confuses the reader and distracts from the main point.

- p7 line 2: It would be helpful to start this section by explicitly explaining purpose of this step to relate the snowpack test data to one of the explanatory factors in the study (i.e. frequency of triggering spots)

- Sect 3.2: This explanation of the bootstrapping method (and the accompanying Fig. 2) are very clear, concise, and effective!

- p7 line 15: What effect does an equal number of samples for each rating have considering there are likely a higher proportion of days with ratings of 2 and 3. The sample of 10,000 will likely have a skewed number of unstable tests from high danger days. Does this impact the interpretation of the results?

- p9 line 28: Slightly confusing, perhaps add ". . . distribution of observed data for all days at a given danger level represent. . ."

- p10 line 1: Consider different verb than "complemented"

- Sect. 4.1.2: This section has many references to appendix figures, which disrupts the flow because the reader is compelled to flip back and forth to the appendix. The confusion could be reduced by introducing Fig. 4 earlier, which clearly shows the most relevant results, then followed by more discussion about the sensitivities to sample size, etc that reference the appendix figures.

- p12 line 10: Are these proportions discussed later? They seem meaningful for interpreting stability test results (e.g. even dangerous days have relatively few sites with very poor stability).

- p14 lines 2-9: This is an example of where the comparison between countries seems like a secondary discussion point compared to reporting the main patterns between avalanche size and danger.

- p15 line 9: In this list the percentages reported in brackets could be misinterpreted as proportion of locations with very poor stability. Perhaps the first reported percentage could explain what the percentage means, e.g. "(53% of sample)".

- Fig. 6-8: Good use of figures with a consistent layout showing the lookup table and the supporting data. The idea that Fig 7 and 8 have the exact same matrix structure as Fig 6 wasn't fully clear on the first read, so could perhaps be explained more explicitly in the text.

- p20 line 17: "while observations of natural or artificial..."

- p20 line 27: Captured "slope stability" or "regional danger"?

- Sect 5.3.1: Another consideration when comparing with existing methods is the CMAH assesses the frequency of trigger spots for each avalanche problem rather than snowpack as a whole as done in the EAWS matrix. This may make it easier to answer questions about the frequency of unstable locations for a specific problem type, but could make it more difficult when combining avalanche problems into an overall danger rating. Just an additional thing to consider when discussing how we can better assess the spatial frequency of instabilities.

- p24 lines 5-9: An updated citation with more comprehensive analysis is Clark (2019), where the influence of many factors on danger ratings are explored (size, likelihood, problem type, region, vegetation band, etc.). The importance of "likelihood" in Clark (2019) still agrees with the main findings in this study.

**References**

• Clark, T.: Exploring the Link between the Conceptual Model of Avalanche Hazard and the North American Public Avalanche Danger Scale, Simon Fraser University, MSc Thesis, http://www.avalancheresearch.ca/pubs/2019$_mrm_clark$, 2019.

• Statham, G., Haegeli, P., Greene, E., Birkeland, K., Israelson, C., Tremper, B., Stethem, C., McMahon, B., White, B., and Kelly, J.: A conceptual model of avalanche hazard, Natural Hazards, 90, 663 − 691, doi:10.1007/s11069-017-3070-5, 2018.
* * *
**TCD**

---

## Referee Comment (RC2) · Karl W. Birkeland (Referee) · 24 Apr 2020

This paper utilizes a data driven approach to look at the relative contributions of snow-pack stability, the frequency distribution of snowpack stability, and avalanche size in assessing avalanche danger. The paper provides a novel analysis of a unique dataset to generate interesting results, and it will be a nice contribution to the literature. I believe the paper should be published after addressing several points.

First, the title could be worded more succinctly and less ambiguously. I might sug-

gest something along the lines of "The importance of snowpack stability, the frequency distribution of snowpack stability, and avalanche size in assessing avalanche danger". However, the authors might have some other title they prefer. In particular I think they could omit "a data-driven approach" since that can be emphasized in the abstract and the text. Also, in the title and in several places in the paper they write "…snowpack stability, its frequency distribution, and avalanche size…". I personally find this to be a bit awkward and ambiguous with the use of the term "its". Even though it is slightly longer and involves more words, I think saying "…snowpack stability, the frequency distribution of snowpack stability, and avalanche size…" states what the authors are trying to say more clearly.

Second, my main criticism of the paper relates to the conclusion by the authors that "avalanche size only has a rather minor influence on the danger level" (bottom of p. 23). Perhaps this is just from the author's choice of words, but in my opinion the data and Figures in the paper do not show a "rather minor influence". Instead, they show an influence that may be less than that of snow stability or frequency, but one that is still clearly evident. An example is in Figure 6 where no matter which letter you get from the combination of stability and frequency on the left side of the Figure, when you go to the right side of the Figure you can see that with all the letters you see an increase in the avalanche danger as the largest avalanche size increases. This is also clearly shown in Figure 8, where going from left to right in the Figure we can see that the proportion of higher danger levels increases as the avalanche size increases.

Another example of the influence of avalanche size can be seen in Figure 5. It is true, as the authors state in the Conclusions on p. 24, that "the largest avalanche size – used by itself – had comparably little discriminating power at 1-Low to 3-Considerable". However, while that might be strictly true for "the largest avalanche size", Figure 5 shows that the distribution of avalanche size – particularly of the largest avalanche (Figure 5b) – clearly does play into avalanche danger. The frequency distributions visibly tend toward larger avalanches at higher danger levels, with the proportion of size

3 and 4 avalanches increasing while the proportion of size 1 avalanches decreases.

I would tend to disagree with the statement on p. 14, line 10-11, that Figure 5b shows "rather similar size distributions at 1-Low and 2-Moderate". Comparing the two, we can see a sizable decrease in size 1 avalanches and an almost doubling in the number of size 3 avalanches between Low and Moderate.

Given the data presented in the paper, I would argue that the authors should better acknowledge that avalanche size does indeed have an influence on avalanche danger, and is not a "rather minor influence" (as stated on p. 23). I think they could still make an argument that snow stability, and the frequency of snow stability might well have a larger influence on avalanche danger, but avalanche size is also an important part of the avalanche danger assessment process. I would therefore encourage them to revisit various parts of the manuscript where avalanche size is discussed and better acknowledge the influence of size on avalanche danger.

Below are some other suggestions and typographical errors that I believe the authors should address:

- p. 1, line 2, delete "the"

- p. 1, line 4, remove the two commas

- p. 2, line 16, replace "weakest" with "the most unstable" because weakest could simply be a weak snowpack that has no slab and is therefore not unstable.

- p. 2, line 23, what does the "(?)" refer to? Were the authors going to put a reference in there or ??

- p. 3, line 2, spell out EAWS completely the first time it is introduced in the text and then refer to it as EAWS afterwards.

- p. 3, line 7, remove the two commas and replace "work" with "works"

- p. 3, line 11, replace "but" with "and"

- p. 3, line 12, replace "And" with "and"

- p. 3, line 13, delete "does" and change "describe" to "describes"

- p. 3, line 24, delete the comma and remove the apostrophe from "biases"

- p. 3, line 25, delete "The target variable" and "we want to describe the three factors with"

- p. 4, line 17, would the authors like to include Foehn, 1987 in addition to Schweizer, 2002 to the RB reference?

- p. 5, line 1, replace "comparably" with "relatively"

- p. 6, line 1-3. It would be nice if the authors would explain why they removed the upper and lower 2.5% of the avalanche data. I am guessing they did this to filter out possible errors with the extremes or something along those lines? In any event, a single sentence explaining why this was done would be helpful.

- p. 7, line 5. The authors state that they are assuming that "different days with the same danger level exhibit similar stability distributions". I think they probably have to assume this to continue with their analyses. However, although I don't have any concrete data to support this, I feel like stability distributions can certainly vary between days that have the same danger level. This is somewhat built into the Conceptual Model of Avalanche Hazard by the inclusion of "uncertainty" and relates to how large an oval a person might put on the probability/size graph of the CMAH when selecting a danger level. It seems to me that the largest variations in stability distributions fall under "3 - Moderate" and "4 - Considerable" danger levels. For example, sometimes under 4 – Considerable you might have a distribution that is more spread out with the possibility of triggering a larger avalanche, while another time you might have a narrower spread of values, but the size of avalanche expected might be smaller. Both of these could have the same avalanche danger level, but the distribution of stability would vary. I don't think the authors have to make big changes to this paper, but I do think they

should acknowledge that this assumption they are making might not always be valid.

- p. 7, line 18, sentence is a bit awkward and confusing. I would change it to read: "Since nature is not as discrete as the danger levels suggest, we wanted both some overlap between our sampled stability distributions and a reasonably high resolution of our statistic."

- p. 9, line 12, replace "maximising" with "maximizing"

- p. 11, Figure 3. This is an interesting and important Figure. One limitation that is noted in the text and also in the figure is the very small N for "4-High" (approximately two orders of magnitude smaller than for 2-Moderate or 3-Considerable). To further emphasize this, the authors could consider stating something related to this in the Figure caption, possibly something like "Note the small N for 4-High for both tests", or, even better, you could write "Note the N for 4-High is small and is approximately two orders of magnitude less than the N for 2-Moderate or 3-Considerable".

- p. 12, line 11, delete the first "of" in the line.

- p. 14, line 19, delete "It is of"

- p. 19, line 4. I have seen this under representation of smaller avalanches in most datasets related to ski area snow safety staff in the United States. This isn't written down in too many places, but we do discuss this somewhat in Birkeland and Landry, 2002 (Power-laws and snow avalanches. Geophysical Research Letters 29(11), 49-1 to 49-3).

- p. 19, line 6. Replace "As" with "Since" and insert "instead" between "focused" and "on".

- p. 19, line 8, delete comma

- p. 19, line 9, replace "weak" with "unstable". I believe the authors are talking about an "unstable" snowpack here and not necessarily one that is just structurally weak,

correct?

- p. 20, line 23 and 25 (and probably elsewhere in the manuscript), replace "in prep." with "under review".

- p. 21, line 23, delete "It is of" and replace "was" with "were"

- p. 21, line 26, replace "," with "." prior to "For instance,"

- p. 21, line 28, replace "Schweizer et al. (2003) s" with "Schweizer et al.'s (2003)"

- p. 22, line 25 and 27. The authors refer to the correlations being "strong" or "moderate". What do you mean by this? Are they statistically significant or not? You might want to state whether they are significant and list a p-value. When I refer back to Section 4.1.2 as is suggested on line 27, I believe the authors are referring to p. 12, line 5-8. Is this correct? Here it states that – even with an N = 10 - the correlation is highly significant (p < 0.001).

- p. 23, line 5. What does the "(?)" refer to? Are the authors planning on adding a reference here?

- p. 30, delete "and tables" from the title of Appendix 2 since this appendix has only figures.

- p. 31, in the caption for Figure B1, replace "Fig.s" with "Figs."

- p. 34, Figure E1, for the top right part of the Figure (all avalanches for Switzerland), add "(SWI)" after "all avalanches" to be consistent with the other headers. Also, add the percent number above the bar for size 1 avalanches under Low to match the other graphs in this Figure.

Again, I believe that this is a good paper and, with some relatively minor changes, will be ready for publication. If the authors have any questions about these comments, they should feel free to email me at karl.birkeland@usda.gov.

Karl Birkeland

---

## Author Comment (AC1) · 9 May 2020

**Reply to reviewer comments Karl Birkeland**

Frank Techel

*Correspondence to:* Frank Techel (techel@slf.ch)

Dear Karl

thank you for your detailed and very helpful review of our manuscript. We greatly appreciate the time and effort you put into reviewing our manuscript.

Please find below our responses (in blue) to your comments *(in italics)*.

*This paper utilizes a data driven approach to look at the relative contributions of snowpack stability, the frequency distribution of snowpack stability, and avalanche size in assessing avalanche danger. The paper provides a novel analysis of a unique dataset to generate interesting results, and it will be a nice contribution to the literature. I believe the paper should be published after addressing several points.*

*First, the title could be worded more succinctly and less ambiguously. I might suggest something along the lines of "The importance of snowpack stability, the frequency distribution of snowpack stability, and avalanche size in assessing avalanche danger". However, the authors might have some other title they prefer. In particular I think they could omit "a data-driven approach" since that can be emphasized in the abstract and the text. Also, in the title and in several places in the paper they write «snowpack stability, its frequency distribution, and avalanche size:» . I personally find this to be a bit awkward and ambiguous with the use of the term "its". Even though it is slightly longer and involves more words, I think saying «snowpack stability, the frequency distribution of snowpack stability, and avalanche size» states what the authors are trying to say more clearly.*

Thank you for this suggestion. We will change the title to «On the importance of snowpack stability, the frequency distribution of snowpack stability, and avalanche size in assessing the avalanche danger level» . Within the text, we will change accordingly where necessary.

*Second, my main criticism of the paper relates to the conclusion by the authors that «avalanche size only has a rather minor influence on the danger level» (bottom of p. 23). Perhaps this is just from the author's choice of words, but in my opinion the data and Figures in the paper do not show a "rather minor influence". Instead, they show an influence that may be less than that of snow stability or frequency, but one that is still clearly evident. An example is in Figure 6 where no matter which letter you get from the combination of stability and frequency on the left side of the Figure, when you go to the right side of the Figure you can see that with all the letters you see an increase in the avalanche danger as the*

*largest avalanche size increases. This is also clearly shown in Figure 8, where going from left to right in the Figure we can see that the proportion of higher danger levels increases as the avalanche size increases. Another example of the influence of avalanche size can be seen in Figure 5. It is true, as the authors state in the Conclusions on p. 24, that "the largest avalanche size – used by itself – had comparably little discriminating power at 1-Low to 3-Considerable". However, while that might be strictly true for "the largest avalanche size", Figure 5 shows that the distribution of avalanche size – particularly of the largest avalanche (Figure 5b) – clearly does play into avalanche danger. The frequency distributions visibly tend toward larger avalanches at higher danger levels, with the proportion of size 3 and 4 avalanches increasing while the proportion of size 1 avalanches decreases. I would tend to disagree with the statement on p. 14, line 10-11, that Figure 5b shows "rather similar size distributions at 1-Low and 2-Moderate". Comparing the two, we can see a sizable decrease in size 1 avalanches and an almost doubling in the number of size 3 avalanches between Low and Moderate. Given the data presented in the paper, I would argue that the authors should better acknowledge that avalanche size does indeed have an influence on avalanche danger, and is not a "rather minor influence" (as stated on p. 23). I think they could still make an argument that snow stability, and the frequency of snow stability might well have a larger influence on avalanche danger, but avalanche size is also an important part of the avalanche danger assessment process. I would therefore encourage them to revisit various parts of the manuscript where avalanche size is discussed and better acknowledge the influence of size on avalanche danger.*

Currently, we describe the findings shown in Fig. 5b with (p. 14 l. 10-14): «Considering the size of the largest reported avalanche per day and warning region showed again rather similar size distributions at 1-Low and 2-Moderate (Fig. 5b). The median largest avalanche per day and region was size 2 for 1-Low to 3-Considerable, except at 4-High with size 3. However, the proportion of days when size 1 avalanches were the largest observed avalanche decreased considerably with increasing danger level, while the proportion of days with at least one size 3 or size 4 avalanche increased monotonically. At 4-High, more than 75% of the days had at least one avalanche of size 3 or 4 recorded.» The statement «rather similar size distributions» is probably a bit ambiguous. Otherwise, this description corresponds to what is shown in the figure. We intend to rephrase to something like: "Considering the size of the largest reported avalanche per day and warning region showed that the largest avalanche per day and region was most frequently size 2 for 1-Low and 2-Moderate, a mix of size 2 and size 3 at 3-Considerable, and size 3 at 4-High (Fig. 5b). However, the proportion of days when size 1 avalanches were the largest observed avalanche decreased significantly with increasing danger level (p < 0.001), while the proportion of days with at least one size 3 or size 4 avalanche increased significantly (p < 0.001). At 4-High, more than 75% of the days had at least one avalanche of size 3 or 4 recorded." Regarding Figures 6 and 8: We agree, row-wise there is a slight increase in the proportion of higher danger levels with increasing avalanche size. However, if cells are ignored which have very few cases (in Fig. 6 cells with less than 1% of the data are highlighted, in Fig. 8 these are the cells with N < 100), then only rather minor differences can be seen. This contrasts to reading the figures column-wise. The danger level changes almost always from 1-Low in row E to 3-Considerable or 4-High in rows A and B. Hence, we believe the statement, which we make on p. 23 l. 33 that «avalanche size only has a rather minor influence on the danger level», is essentially true. However, we propose to rephrase along the line: «Even though there was a shift towards

more frequently observed larger avalanches with increasing danger level, the danger level was primarily influenced by the cell describing stability. [could add] Avalanche size (apparently) influences the danger level only if avalanches are small (size=1) or very large (size≥4).«- Please note that we intend to restructure the Results section, following the suggestions by referee Simon Horton. This means, that we will probably present the Swiss data first, and will therefore phrase the sentence again slightly different.

Thank you for the following suggestions and pointing out typographical errors, which we will address when revising the manuscript. Please find below comments on some of the points. However, we will provide a point-by-point reply together with our revised manuscript.

*Below are some other suggestions and typographical errors that I believe the authors should address:*

- *p. 1, line 2, delete "the"*

- *p. 1, line 4, remove the two commas*

- *p. 2, line 16, replace "weakest" with "the most unstable" because weakest could simply be a weak snowpack that has no slab and is therefore not unstable.* - We intend to use as an alternative term, probably something along the line "lowest stability" rather than "weak" or "the most unstable".

- *p. 2, line 23, what does the "(?)" refer to? Were the authors going to put a reference in there or ??*

- *p. 3, line 2, spell out EAWS completely the first time it is introduced in the text and then refer to it as EAWS afterwards.*

- *p. 3, line 7, remove the two commas and replace "work" with "works"*

- *p. 3, line 11, replace "but" with "and"*

- *p. 3, line 12, replace "And" with "and"*

- *p. 3, line 13, delete "does" and change "describe" to "describes"*

- *p. 3, line 24, delete the comma and remove the apostrophe from "biases"*

- *p. 3, line 25, delete "The target variable" and "we want to describe the three factors with"*

- *p. 4, line 17, would the authors like to include (Föhn, 1987) in addition to (Schweizer, 2002) to the RB reference?*

- *p. 5, line 1, replace "comparably" with "relatively"*

– *p. 6, line 1-3. It would be nice if the authors would explain why they removed the upper and lower 2.5% of the avalanche data. I am guessing they did this to filter out possible errors with the extremes or something along those lines? In any event, a single sentence explaining why this was done would be helpful.* - We will provide an explanation.

– *p. 7, line 5. The authors state that they are assuming that "different days with the same danger level exhibit similar stability distributions". I think they probably have to assume this to continue with their analyses. However, although I don't have any concrete data to support this, I feel like stability distributions can certainly vary between days that have the same danger level. This is somewhat built into the Conceptual Model of Avalanche Hazard by the inclusion of "uncertainty" and relates to how large an oval a person might put on the probability/size graph of the CMAH when selecting a danger level. It seems to me that the largest variations in stability distributions fall under "3 - Moderate" and "4 - Considerable" danger levels. For example, sometimes under 4 – Considerable you might have a distribution that is more spread out with the possibility of triggering a larger avalanche, while another time you might have a narrower spread of values, but the size of avalanche expected might be smaller. Both of these could have the same avalanche danger level, but the distribution of stability would vary. I don't think the authors have to make big changes to this paper, but I do think they should acknowledge that this assumption they are making might not always be valid.* - We agree, there is not one typical stability distribution for each danger level, but rather a typical range of distributions. This is essentially what we got when we applied the bootstrap-sampling approach (e.g. Figure C1c in the Appendix). At 3-Considerable, there were distributions with a proportion of 2% very poor and 25% good stability tests, but also some where these proportions were more than 20% very poor and 5% good. - We will rephrase accordingly.

– *p. 7, line 18, sentence is a bit awkward and confusing. I would change it to read: "Since nature is not as discrete as the danger levels suggest, we wanted both some overlap between our sampled stability distributions and a reasonably high resolution of our statistic."*

– *p. 9, line 12, replace "maximising" with "maximizing"*

– *p. 11, Figure 3. This is an interesting and important Figure. One limitation that is noted in the text and also in the figure is the very small N for "4-High" (approximately two orders of magnitude smaller than for 2-Moderate or 3-Considerable). To further emphasize this, the authors could consider stating something related to this in the Figure caption, possibly something like "Note the small N for 4-High for both tests", or, even better, you could write "Note the N for 4-High is small and is approximately two orders of magnitude less than the N for 2-Moderate or 3-Considerable".* - We will add a note in this regard.

– *p. 12, line 11, delete the first "of" in the line.*

– *p. 14, line 19, delete "It is of"*

- *p. 19, line 4. I have seen this under representation of smaller avalanches in most datasets related to ski area snow safety staff in the United States. This isn't written down in too many places, but we do discuss this somewhat in (Birkeland and Landry, 2002) (Power-laws and snow avalanches. Geophysical Research Letters 29(11), 49-1 to 49-3).* Thank you for pointing this out. This supports the findings shown in 5a, and the literature that we have cited in the Discussion section (p. 18 bottom and p. 19).

- *p. 19, line 6. Replace "As" with "Since" and insert "instead" between "focused" and "on".*

- *p. 19, line 8, delete comma*

- *p. 19, line 9, replace "weak" with "unstable". I believe the authors are talking about an "unstable" snowpack here and not necessarily one that is just structurally weak, correct?*

- *p. 20, line 23 and 25 (and probably elsewhere in the manuscript), replace "in prep." with "under review".*

- *p. 21, line 23, delete "It is of" and replace "was" with "were"*

- *p. 21, line 26, replace "," with "." prior to "For instance,"*

- *p. 21, line 28, replace "Schweizer et al. (2003) s" with "Schweizer et al.'s (2003)"*

- *p. 22, line 25 and 27. The authors refer to the correlations being "strong" or "moderate". What do you mean by this? Are they statistically significant or not? You might want to state whether they are significant and list a p-value. When I refer back to Section 4.1.2 as is suggested on line 27, I believe the authors are referring to p. 12, line 5-8. Is this correct? Here it states that – even with an N = 10 - the correlation is highly significant (p < 0.001).* - Yes, these terms refer to the way Spearman rank-order correlations are interpreted. Both were indeed highly significant. - We will rephrase.

- *p. 23, line 5. What does the "(?)" refer to? Are the authors planning on adding a reference here?*

- *p. 30, delete "and tables" from the title of Appendix 2 since this appendix has only figures.*

- *p. 31, in the caption for Figure B1, replace "Fig.s" with "Figs."*

- *p. 34, Figure E1, for the top right part of the Figure (all avalanches for Switzerland), add "(SWI)" after "all avalanches" to be consistent with the other headers. Also, add the percent number above the bar for size 1 avalanches under Low to match the other graphs in this Figure.*

Frank Techel, on behalf of all co-authors

**References**

Birkeland, K. and Landry, C.: Power-laws and snow avalanches, Geophysical Research Letters, 29, doi:10.1029/2001GL014623, 2002.

Föhn, P.: The rutschblock as a practical tool for slope stability evaluation, IAHS Publ., 162, 223–228, 1987.

Schweizer, J.: The Rutschblock test - procedure and application in Switzerland, The Avalanche Review, 20, 14–15, 2002.

---

## Author Comment (AC2) · 9 May 2020

**Reply to reviewer comments by Simon Horton**

Frank Techel

*Correspondence to:* Frank Techel (techel@slf.ch)

Dear Simon

thank you very much for your very detailed and helpful review of our manuscript. We greatly appreciate the time and effort you put into this review.

Please find below our reply (in blue) to your comments *(in italics)*.

*General comments*
*The paper addresses the subjective nature of avalanche hazard ratings by exploring a large dataset of hazard ratings and field observations. Since hazard ratings are defined by subjective terms, this study presents a unique approach to quantifying some of these terms. This could improve the consistency of hazard assessments and risk communication, which is an important contribution. The scientific methods applied in the paper are rigorous, and the conclusions drawn from the results are appropriate and interesting.*
Thank you for this positive feedback to our manuscript.

*My main comment is the key results of the study could be more clearly communicated and emphasized. I found there were some additional data and results that distracted from the key messages, and as a result I needed to read certain sections twice to properly understand the relevance. I think this could be easily addressed by restructuring and shortening some sections. My other main comment is I think more discussion about the application of the results (e.g. improved forecasting methods, danger rating definitions) would make the contributions clearer to the reader.*

*Specific comments*
*Relevance of additional data sets: The key conclusions of the study appear to come from the Swiss Rutschblock test and avalanche data, however while reading the results there are numerous references to patterns between the Swiss/Norwegian and RB/ECT data. I found this distracting from the main research question about the contributing factors to danger ratings. I would consider restructuring some of the sections so the main research question is addressed first, and then perhaps distinct sub-sections discussing how the core results differ between SWI/NOR and between RB/ECT results. There are also quite a few Appendix figures with these additional data sets which disrupts the flow while reading the results.* Thank you for pointing this out. We agree, the structure of the manuscript should be improved to make it easier for the reader to

follow the line of argumentation. We intend to restructure the Results section (Sect. 4) in a way that we first present the results using the Swiss data only, showing relevant figures and tables in the main part of the manuscript. Results relating to the methodology (bootstrap-sampling and classification approach), which we consider equally important to highlight strengths and limitations of the approach taken, we will move to a separate subsection. Other results, like the comparison RB to ECT, or comparing Swiss to Norwegian data, will be moved to a separate subsection with additional figures in the Appendix. In the Discussion section, however, we will keep the subsections where we discuss the limitations of the data used and the methodology, as we consider these relevant to understand the limitations of the study. Readers who are primarily interested in the findings of the study relating to the research questions, should be able to ignore these subsections easily.

*Applications/next steps: I found the discussion focused too much on the limitations of the study rather than focusing on an in-depth discussion of how the results relate to the main research question. I would suggest shortening the limitations and adding more discussion about how these results could be applied to improve danger ratings. Linkages to existing hazard assessment frameworks (e.g. ADAM, CMAH) are discussed, but I would have found it interesting to read more about how the results could improve or unify the existing frameworks. For example, the stability matrix in Fig. 6 has many parallels with the likelihood matrix from the CMAH (Statham et al. (2018), Fig. 2), the Bavarian matrix, and ADAM. While these are discussed, I think the strong quantitative data in this study are well positioned to make an informed critique on existing methods and more suggestions for future directions.*

*Applying stability distributions: While the bootstrap sampling method is appropriate to derive stability distributions and define classes, it is somewhat theoretical and not clear how the derived classes could be applied in practice. Can these distributions help us understand more about stability conditions for a given hazard level? For example, from Fig. 3 could we assume that roughly 17% of slopes are unstable at 3-Considerable and 38% of slopes are unstable at stable at 4-High? I understand the challenges with making that inference, however, I wonder if the numeric nature of the study could help give some additional meaning to terms like few, several, and many. And although the theoretical meaning of these terms are clearly defined, how can forecasters actually assess whether the frequency of unstable locations is few/several/many? -* Concerning these two points (application-next steps / applying stability distributions): With this study, it is our intention to provide the avalanche forecasting community with results regarding the contributing factors of avalanches grounded in data and robust methodologies. Primarily, we hope these findings will stimulate the discussion in the avalanche forecaster community when revising key terms and their definitions (a current project in the working group of the European Avalanche Warning Services) and the Avalanche Danger Scale. - We intend to take up some of the quantitative key results again, like the proportion of very poor tests at 4-High or the quantitative ranges describing the four frequency classes, and put these in relation with other studies (e.g. Thumlert et al., 2020)

*Methods: The beginning of each sub-section could use a bit more context about how that step is relevant to exploring the link between danger ratings and a contributing factor. -* We will add more context at the beginning of each section.

Figures: The figures are clear, legible, and support the main messages of the study well.

Thank you for pointing out typos and other improvements in your technical comments. We will address them when revising the manuscript and provide a detailed point-by-point reply when submitting a revised version of the manuscript. Please find our feedback regarding the more important points below.

*Technical comments*

- *p1 line 7: Although less precise, saying "frequency of unstable locations" may be simpler to understand when reading the abstract only*

- *p1 line 13-14: Consider adding "simulated stability distributions" (the snowpack distribution isn't simulated)*

- *p2 line7: Preferable to use consistent terminology from the list of key factors, i.e. "probability of avalanche release" instead of "release (or triggering) probability"*

- *p2 line 13: Similar to above, starting the paragraph by repeating the term "frequency and location of triggering spots" would make it clearer the paragraph ties back to the list of key factors*

- *p2 line 23: missing citation*

- *p2 line 24: According to the CMAH spatial distribution also considers spatial density. Statham et al. (2018) "Spatial distribution considers the spatial density and distribution of an avalanche problem and the ease of finding evidence to support or refute its presence."*

- *Table 1: The "data from" column heading isn't clear if the data is from just a single season or all seasons up to 2018/19 (as explained in footnote). Consider a more precise heading or list season ranges in the table (e.g. 2002-2019)*

- *p4 line 2-4: These two sentences aren't necessary, as they are discussed below.*

- *p6 line 4-5: Please be consistent with order of reporting SWI and NOR data, in this sentence NOR is described first.*

- *p6 line 8: It would be helpful to start this section by explicitly explaining the purpose of this step is to relate the snowpack test data to one of the explanatory factors in the study (i.e. probability of avalanche release) - We will add some explanation in this regard.*

- *p6 lines 19-26: This is an example of how the addition of ECT data confuses the reader and distracts from the main point.*

– *p7 line 2: It would be helpful to start this section by explicitly explaining purpose of this step to relate the snowpack test data to one of the explanatory factors in the study (i.e. frequency of triggering spots)*

– *Sect 3.2: This explanation of the bootstrapping method (and the accompanying Fig. 2) are very clear, concise, and effective!*

– *p7 line 15: What effect does an equal number of samples for each rating have considering there are likely a higher proportion of days with ratings of 2 and 3. The sample of 10,000 will likely have a skewed number of unstable tests from high danger days. Does this impact the interpretation of the results?* - An equal number of samples for each danger level is important, when the danger level for each combination is sought. For instance, if only 1% of the samples would have been 4-High, the danger matrix in Figure 6 would essentially never show a 4-High, as 3-Considerable would dominate these cells due to their larger weight. The definition of the class thresholds changes little, as the median proportion of very poor tests $VP_{med}$) drives them. When using a typical distribution of danger levels forecast in Switzerland instead (1-Low to 4-High, 18%, 43%, 36%, 2%, respectively), the variable which defines the class intervals $VP_{med}$ is the same with 0.08. – We will add an explanation in that respect.

– *p9 line 28: Slightly confusing, perhaps add "... distribution of observed data for all days at a given danger level represent..."*

– *p10 line 1: Consider different verb than "complemented"*

– *Sect. 4.1.2: This section has many references to appendix figures, which disrupts the flow because the reader is compelled to flip back and forth to the appendix. The confusion could be reduced by introducing Fig. 4 earlier, which clearly shows the most relevant results, then followed by more discussion about the sensitivities to sample size, etc that reference the appendix figures.* - As pointed out before, we intend to restructure this section to make it easier for the reader to follow the line of argumentation.

– *p12 line 10: Are these proportions discussed later? They seem meaningful for interpreting stability test results (e.g. even dangerous days have relatively few sites with very poor stability).* - No, we did not discuss them later. They are the central tendency values for the four classes. - We will take them up at a later stage to highlight the quantitative meaning of the classes.

– *p14 lines 2-9: This is an example of where the comparison between countries seems like a secondary discussion point compared to reporting the main patterns between avalanche size and danger.* - These will be moved to a separate subsection.

– *p15 line 9: In this list the percentages reported in brackets could be misinterpreted as proportion of locations with very poor stability. Perhaps the first reported percentage could explain what the percentage means, e.g. "(53% of sample)".*

– *Fig. 6-8: Good use of figures with a consistent layout showing the lookup table and the supporting data. The idea that Fig 7 and 8 have the exact same matrix structure as Fig 6 wasn't fully clear on the first read, so could perhaps be explained more explicitly in the text.*

– *p20 line 17: "while observations of natural or artificial..."*

– *p20 line 27: Captured "slope stability" or "regional danger"?*

– *Sect 5.3.1: Another consideration when comparing with existing methods is the CMAH assesses the frequency of trigger spots for each avalanche problem rather than snowpack as a whole as done in the EAWS matrix. This may make it easier to answer questions about the frequency of unstable locations for a specific problem type but could make it more difficult when combining avalanche problems into an overall danger rating. Just an additional thing to consider when discussing how we can better assess the spatial frequency of instabilities. -* We agree that assessing the spatial frequency of instabilities in an actual situation is certainly not an easy task. Focussing on a specific avalanche problem may indeed allow to make a process-based guess. Still, as you state, the final compilation into one danger level is not straightforward – unless you opt for the worst combination across the various problems.

– *p24 lines 5-9: An updated citation with more comprehensive analysis is Clark (2019), where the influence of many factors on danger ratings are explored (size, likelihood, problem type, region, vegetation band, etc.). The importance of "likelihood" in Clark (2019) still agrees with the main findings in this study.*

**References**

Clark, T.: Exploring the link between the Conceptual Model of Avalanche Hazard and the North American Public Avalanche Danger Scale, Master's thesis, Simon Fraser University, 115 p., 2019.

Statham, G., Haegeli, P., Greene, E., Birkeland, K., Israelson, C., Tremper, B., Stethem, C., McMahon, B., White, B., and Kelly, J.: A conceptual model of avalanche hazard, Natural Hazards, 90, 663 – 691, doi:10.1007/s11069-017-3070-5, 2018.

Thumlert, S., Statham, G., and Jamieson, B.: The likelihood scale in avalanche forecasting, The Avalanche Review, 38, 31–33, 2020.

---

## Author Response (AR1)

**Dear Editor, dear Simon Horton and Karl Birkeland**

Thank you very much for your detailed and constructive review of our, admittedly, rather complex manuscript.
The revised manuscript is certainly still rather complex, due to the description and evaluation of the methodology needed to simulate snow stability distributions and classify the frequency of the most unstable locations. However, we hope that after restructuring the Result section and integrating (or removing) supplementary information previously shown in the Appendix, the manuscript is now easier to read. We did, however, leave the paragraphs in the manuscript, where we present and discuss the strength and limitations of the approach taken. We believe, readers which are not interested in this can now easily skip these sections and focus on the main results, while those interested in the limitations will find these addressed in the manuscript.

Please find following
- A list indicating the major changes
- A point-by-point reply to the reviews
- The manuscript showing all the track changes.

**Major changes**

- We took up the recommendations by reviewer #1 and restructured large parts of the Result section (Sect 4). However, the actual results (or their interpretation) were not affected by this restructuring of the manuscript. Figures shown in the Appendix were either integrated in the new Results section, or they were removed. In the track-changes document, the entire Sect. 4 is marked, as we had to restructure and rephrase large parts of this section.
- Sect. 4.1 – 4.4 now show the findings based on the Swiss data. Sect. 4.5 presents the ECT data and the data from Norway and compares them to the respective Swiss or Rutschblock data. Sect. 4.6 presents results related to the methodology of bootstrap sampling and frequency classes.
- Fig. 4 is new. It shows the distribution of the frequency classes of very poor stability for the four danger levels.
- For stability classification of ECT and RB, we used a simpler depth criterion (p6 l17-19) than in the original version (p6 l28-31). We followed the approach taken by Techel et al. (*On snow stability interpretation of Extended Column Test results*, NHESS, accepted). Very marginal changes in the proportions observed for RB and ECT (Fig. 3) resulted.

**Review by reviewer #1 Simon Horton**

| Original manuscript (reviewer comment) | Revised manuscript (changes made) |
|---|---|
| Relevance of additional data sets: The key conclusions of the study appear to come from the Swiss Rutschblock test and avalanche data, however while reading the results there are numerous references to patterns between the Swiss/Norwegian and RB/ECT data. I found this distracting from the main research question about the contributing factors to danger ratings. I would consider restructuring some of the sections so the main research question is addressed first, and then perhaps distinct sub- | We have completely restructured the Results section, to address this point. We now show first all the results based on Swiss data (Sect. 4.1 – 4.4), we then compare these in a second step with Norwegian data and/or ECT (Sect. 4.5). Results, which were related more to the methodology rather than linking the observations to avalanche hazard, have been moved to a new Sect. 4.6 together with relevant figures. |

| sections discussing how the core results differ between SWI/NOR and between RB/ECT results. There are also quite a few Appendix figures with these additional data sets which disrupts the flow while reading the results. | We hope this restructure helps the reader to distinguish more easily between the results relating to the research questions, and those that compare to other data sets or that present additional information regarding the methodology. Furthermore, together with this restructure, we have moved the relevant figures from the Appendix to the main part of the manuscript. |
|---|---|
| Methods: The beginning of each sub-section could use a bit more context about how that step is relevant to exploring the link between danger ratings and a contributing factor. | We added a sentence in that regard in the subsections: p6 l3, p6 l21 |
| p1 line 7: Although less precise, saying "frequency of unstable locations" may be simpler to understand when reading the abstract only | p1 l7: According to reviewer #2 changed to *…the frequency distribution of snowpack stability and…* |
| p1 line 13-14: Consider adding "simulated stability distributions" (the snowpack distribution isn't simulated) | P1 l13: changed to *…simulated stability distributions…* |
| p2 line7: Preferable to use consistent terminology from the list of key factors, i.e. "probability of avalanche release" instead of "release (or triggering) probability" | We changed the terminology in the list (p2 l3-5) and repeated the term when starting each paragraph (p2 l7, p2 l13, p2 l28) |
| p2 line 13: Similar to above, starting the paragraph by repeating the term "frequency and location of triggering spots" would make it clearer the paragraph ties back to the list of key factors | |
| p2 line 23: missing citation | P2 l22-23: we added *…(EAWS, 2017).* |
| p2 line 24: According to the CMAH spatial distribution also considers spatial density. Statham et al. 2018: "Spatial distribution considers the spatial density and distribution of an avalanche problem and the ease of finding evidence to support or refute its presence." | P2 l25-27: we changed to *In the CMAH (Statham et al. 2018a), on the other hand, the spatial distribution is related to the spatial density and distribution of an avalanche problem and the ease of finding evidence for it, and is described using the three terms isolated, specific and widespread.* |
| Table 1: The "data from" column heading isn't clear if the data is from just a single season or all seasons up to 2018/19 (as explained in footnote). Consider a more precise heading or list season ranges in the table (e.g. 2002-2019) | We changed the year format to 2002 - 2019 |
| p4 line 2-4: These two sentences aren't necessary, as they are discussed below. | We removed these sentences. |
| p6 line 4-5: Please be consistent with order of reporting SWI and NOR data, in this sentence NOR is described first. | We changed to always introducing SWI data first. |
| p6 line 8: It would be helpful to start this section by explicitly explaining the purpose of this step is to relate the snowpack test data to | P6 l3: We added *Snowpack stability is one of the three contributing factors to avalanche hazard and relates to the probability of avalanche release.* |

| | |
|---|---|
| one of the explanatory factors in the study (i.e. probability of avalanche release) | |
| p6 lines 19-26: This is an example of how the addition of ECT data confuses the reader and distracts from the main point. | P6 l13-16: We shortened this paragraph considerably. |
| p7 line 2: It would be helpful to start this section by explicitly explaining purpose of this step to relate the snowpack test data to one of the explanatory factors in the study (i.e. frequency of triggering spots) | P6 l21: We added *The second factor contributing to avalanche hazard is the frequency of potential triggering locations, or of snowpack stability.* |
| p7 line 15: What effect does an equal number of samples for each rating have considering there are likely a higher proportion of days with ratings of 2 and 3. The sample of 10,000 will likely have a skewed number of unstable tests from high danger days. Does this impact the interpretation of the results? | An equal number of samples for each danger level is important, when the danger level for each combination is sought. For instance, if only 1% of the samples would have been 4-High, the danger matrix in Figure 6 would essentially never show a 4-High, as 3-Considerable would dominate these cells due to their larger weight. The definition of the class thresholds changes little, as the median proportion of very poor tests ($VP_{med}$) drives them. When using a typical distribution of danger levels forecast in Switzerland instead (1-Low to 4-High, 18%, 43%, 36%, 2%, respectively), the variable which defines the class intervals $VP_{med}$ is the same with 0.08. |
| p9 line 28: Slightly confusing, perhaps add ": : : distribution of observed data for all days at a given danger level represent: : :" | P9 l11-14: We changed to *As we do not have data describing the three factors relating to the same day and region, we used a simulation approach by assuming that the distribution of the observed data represents the typical values and ranges at a specific danger level.* |
| p10 line 1: Consider different verb than "complemented" | P9 l17: we changed to *…we combined the snowpack stability…* |
| Sect. 4.1.2: This section has many references to appendix figures, which disrupts the flow because the reader is compelled to flip back and forth to the appendix. The confusion could be reduced by introducing Fig. 4 earlier, which clearly shows the most relevant results, then followed by more discussion about the sensitivities to sample size, etc that reference the appendix figures. | We completely restructured the Result section (Sect. 4). Now, there are no more references to appendix figures as we have moved all the relevant figures to the Results section. |
| P12 line 10: Are these proportions discussed later? They seem meaningful for interpreting stability test results (e.g. even dangerous days have relatively few sites with very poor stability). | We now discuss these proportions at several locations: P10 l10 – p11l2 and in the Discussion p24 l5-7 |
| p14 lines 2-9: This is an example of where the comparison between countries seems like a | Addressed by restructuring of Results section – new in a separate subsection (Sect. 4.5) |

| Original manuscript (reviewer comment) | Revised manuscript (changes made) |
|---|---|
| secondary discussion point compared to reporting the main patterns between avalanche size and danger. | |
| p15 line 9: In this list the percentages reported in brackets could be misinterpreted as proportion of locations with very poor stability. Perhaps the first reported percentage could explain what the percentage means, e.g. "(53% of sample)". | P14 l9: changed as suggested |
| Fig. 6-8: Good use of figures with a consistent layout showing the lookup table and the supporting data. The idea that Fig 7 and 8 have the exact same matrix structure as Fig 6 wasn't fully clear on the first read, so could perhaps be explained more explicitly in the text. | As suggested, we tried to emphasize that the structure of the figures is the same, in the caption and the text. Additionally, Fig 7 and Fig 8 are now beside each other in Fig 7 (as a and b), thus it should be easier to compare these figures with Fig 6. |
| P20 line 17: "while observations of natural or artificial: : :" | P22 l6-7: changed as suggested |
| p20 line 27: Captured "slope stability" or "regional danger"? | P22 l15-17: we meant slope stability, but the reference to this statement was missing and has now been added *However, as shown by Techel et al. (2020), the most favorable and the most unfavorable RB stability classes captured slope stability better than the respective ECT classes, indicating a lower agreement between slope stability and ECT results compared to the RB.* |
| Sect 5.3.1: Another consideration when comparing with existing methods is the CMAH assesses the frequency of trigger spots for each avalanche problem rather than snowpack as a whole as done in the EAWS matrix. This may make it easier to answer questions about the frequency of unstable locations for a specific problem type but could make it more difficult when combining avalanche problems into an overall danger rating. Just an additional thing to consider when discussing how we can better assess the spatial frequency of instabilities. | We have not taken up this point. |
| p24 lines 5-9: An updated citation with more comprehensive analysis is Clark (2019), where the influence of many factors on danger ratings are explored (size, likelihood, problem type, region, vegetation band, etc.). The importance of "likelihood" in Clark (2019) still agrees with the main findings in this study. | We now make a reference to Clark, 2019 and Clark and Haegeli, 2018 |

Comments by reviewer #2 Karl Birkeland

| Original manuscript (reviewer comment) | Revised manuscript (changes made) |
|---|---|
| First, the title could be worded more succinctly and less ambiguously. I might suggest | We have changed the title to "*On the importance of snowpack stability, the frequency distribution of snowpack stability, and* |

| | |
|---|---|
| something along the lines of "The importance of snowpack stability, the frequency distribution of snowpack stability, and avalanche size in assessing avalanche danger". However, the authors might have some other title they prefer. In particular I think they could omit "a data-driven approach" since that can be emphasized in the abstract and the text. Also, in the title and in several places in the paper they write ": : :snowpack stability, its frequency distribution, and avalanche size: : :". I personally find this to be a bit awkward and ambiguous with the use of the term "its". Even though it is slightly longer and involves more words, I think saying ": : :snowpack stability, the frequency distribution of snowpack stability, and avalanche size: : :" states what the authors are trying to say more clearly. | *avalanche size in assessing the avalanche danger level*". We made similar changes at various locations in the manuscript, for instance:
 P1 l6-7, p7 l21, p9 l10, … |
| Second, my main criticism of the paper relates to the conclusion by the authors that "avalanche size only has a rather minor influence on the danger level" (bottom of p. 23). Perhaps this is just from the author's choice of words, but in my opinion the data and Figures in the paper do not show a "rather minor influence". Instead, they show an influence that may be less than that of snow stability or frequency, but one that is still clearly evident. An example is in Figure 6 where no matter which letter you get from the combination of stability and frequency on the left side of the Figure, when you go to the right side of the Figure you can see that with all the letters you see an increase in the avalanche danger as the largest avalanche size increases. This is also clearly shown in Figure 8, where going from left to right in the Figure we can see that the proportion of higher danger levels increases as the avalanche size increases. Another example of the influence of avalanche size can be seen in Figure 5. It is true, as the authors state in the Conclusions on p. 24, that "the largest avalanche size – used by itself – had comparably little discriminating power at 1-Low to 3-Considerable".  However, while that might be strictly true for "the largest avalanche size", Figure 5 shows that the distribution of avalanche size – particularly of the largest avalanche (Figure 5b) – clearly does play into avalanche danger. The frequency distributions visibly tend toward larger avalanches at higher danger levels, with the proportion of size 3 and | See also our response to this comment. We rephrased at several locations in the manuscript:
 P11 l15 – p16l4: we rephrased in several locations, also because of the restructuring of the Results section
 P25 l12-13:
 *In general, avalanche size had a lesser influence on the danger level, once the cell describing stability has been fixed, as might be anticipated*
 P26 l6-9:
 *Considering the largest observed avalanche size per day and warning region was most relevant to distinguish between 3-Considerable and 4-High (Fig. 5 and Tab. 3). For other situations, the largest avalanche size - when used on its own - had less discriminating power to distinguish between danger levels 1-Low to 3-Considerable compared to the other two factors (the lowest stability class present and the frequency of this class; Fig. 5).* |

| | |
|---|---|
| 4 avalanches increasing while the proportion of size 1 avalanches decreases.

I would tend to disagree with the statement on p. 14, line 10-11, that Figure 5b shows "rather similar size distributions at 1-Low and 2-Moderate". Comparing the two, we can see a sizable decrease in size 1 avalanches and an almost doubling in the number of size 3 avalanches between Low and Moderate.

Given the data presented in the paper, I would argue that the authors should better acknowledge that avalanche size does indeed have an influence on avalanche danger, and is not a "rather minor influence" (as stated on p. 23). I think they could still make an argument that snow stability, and the frequency of snow stability might well have a larger influence on avalanche danger, but avalanche size is also an important part of the avalanche danger assessment process. I would therefore encourage them to revisit various parts of the manuscript where avalanche size is discussed and better acknowledge the influence of size on avalanche danger. | |
| p. 1, line 2, delete "the" | done |
| p. 1, line 4, remove the two commas | done |
| p. 2, line 16, replace "weakest" with "the most unstable" because weakest could | P2 l16: changed to
*…lowest…* |
| p. 2, line 23, what does the "(?)" refer to? Were the authors going to put a reference in there or ?? | P2 l22-23: we added *…(EAWS, 2017).* |
| p. 3, line 2, spell out EAWS completely the first time it is introduced in the text and then refer to it as EAWS afterwards. | done |
| p. 3, line 7, remove the two commas and replace "work" with "works" | done |
| p. 3, line 11, replace "but" with "and" | done |
| p. 3, line 12, replace "And" with "and" | done |
| p. 3, line 13, delete "does" and change "describe" to "describes" | done |
| p. 3, line 25, delete "The target variable" and "we want to describe the three factors with" | done |
| p. 4, line 17, would the authors like to include Foehn, 1987 in addition to Schweizer, 2002 to the RB reference? | We added
*Föhn, 1987* |
| p. 5, line 1, replace "comparably" with "relatively" | done |
| p. 6, line 1-3. It would be nice if the authors would explain why they removed the | P5 l26-27: we added
*…were considered to represent errors in the local estimate of the danger level or of* |

| | |
|---|---|
| upper and lower 2.5% of the avalanche data. I am guessing they did this to filter out possible errors with the extremes or something along those lines? In any event, a single sentence explaining why this was done would be helpful. | *avalanche size. These potentially erroneous data were removed.* |
| p. 7, line 5. The authors state that they are assuming that "different days with the same danger level exhibit similar stability distributions". I think they probably have to assume this to continue with their analyses. However, although I don't have any concrete data to support this, I feel like stability distributions can certainly vary between days that have the same danger level. This is somewhat built into the Conceptual Model of Avalanche Hazard by the inclusion of "uncertainty" and relates to how large an oval a person might put on the probability/size graph of the CMAH when selecting a danger level. It seems to me that the largest variations in stability distributions fall under "3 - Moderate" and "4 - Considerable" danger levels. For example, sometimes under 4 – Considerable you might have a distribution that is more spread out with the possibility of triggering a larger avalanche, while another time you might have a narrower spread of values, but the size of avalanche expected might be smaller. Both of these could have the same avalanche danger level, but the distribution of stability would vary. I don't think the authors have to make big changes to this paper, but I do think they should acknowledge that this assumption they are making might not always be valid. | P7 l2-4: we changed to *Assuming that a single test result is just one sample from the stability distribution on that day and that different days with the same danger level exhibit a range of similar stability distributions, …*

By using the sampling approach, we created a wide range of stability distributions, which we exemplarily describe on p17 l32 – p18 l5 (together with Fig 8c):
*When introducing the bootstrap-sampling approach to create a range of plausible stability distributions (Sect. 3.2), we had to assume that a single stability rating is just one sample from the stability distribution on that day and that different days with the same danger level exhibit a range of similar stability distributions. Referring to Fig. 8c, which shows the proportions of very poor and good stability of the 10,000 simulated distributions with n= 25, it can be noted that indeed a range of typical distributions was obtained for the four danger levels. For instance, at 3-Considerable the range of the simulated distributions was wide: 11% of the samples drawn had ≥ 8% (frequency classes several or many) very poor and ≤ 4% (a few or none) good tests results, while 7% of the samples drawn had ≤ 4% (a few or none) very poor and 24% (many) good tests results.* |
| p. 7, line 18, sentence is a bit awkward and confusing. I would change it to read: "Since nature is not as discrete as the danger levels suggest, we wanted both some overlap between our sampled stability distributions and a reasonably high resolution of | P7 l15-17: done |
| p. 9, line 12, replace "maximising" with "maximizing" | done |
| p. 11, Figure 3. This is an interesting and important Figure. One limitation that is noted in the text and also in the figure is the very small N for "4-High" (approximately two orders of magnitude smaller than for 2-Moderate or 3-Considerable). To further | Fig. 3 and p9 l27-28: we added a remark in this regard. |

| | |
|---|---|
| emphasize this, the authors could consider stating something related to this in the Figure caption, possibly something like "Note the small N for 4-High for both tests", or, even better, you could write "Note the N for 4-High is small and is approximately two orders of magnitude less than the N for 2-Moderate or 3-Considerable". | |
| p. 12, line 11, delete the first "of" in the line. | done |
| p. 14, line 19, delete "It is of" | Sentence removed |
| p. 19, line 4. I have seen this under representation of smaller avalanches in most datasets related to ski area snow safety staff in the United States. This isn't written down in too many places, but we do discuss this somewhat in Birkeland and Landry, 2002 (Power-laws and snow avalanches. Geophysical Research Letters 29(11), 49-1 to 49-3). | P24 l29-31: We rephrased and added references in this regard *This frequency-magnitude relation has also been observed for other natural hazards (e.g. Malamud and Turcotte, 1999), and has been described by power laws for avalanche size distributions (Birkeland and Landry, 2002; Faillettaz et al., 2004).* |
| p. 19, line 6. Replace "As" with "Since" and insert "instead" between "focused" and "on". | done |
| p. 19, line 8, delete comma | done |
| p. 19, line 9, replace "weak" with "unstable". I believe the authors are talking an "unstable" snowpack here and not necessarily one that is just structurally weak, correct? | P21 l3 Changed to low |
| p. 21, line 26, replace "," with "." prior to "For instance," | |
| p. 21, line 28, replace "Schweizer et al. (2003) s" with "Schweizer et al.'s (2003)" | done |
| p. 22, line 25 and 27. The authors refer to the correlations being "strong" or "moderate". What do you mean by this? Are they statistically significant or not? You might want to state whether they are significant and list a p-value. When I refer back to Section 4.1.2 as is suggested on line 27, I believe the authors are referring to p. 12, line 5-8. Is this correct? Here it states that – even with an N = 10 - the correlation is highly significant (p < 0.001). | We removed this part of the Discussion, as it is addressed in the Results section (Sect 4.6.2) p18 l23-24 |
| p. 23, line 5. What does the "(?)" refer to? Are the authors planning on adding a reference here? | We added the reference (EAWS, 2017) |
| p. 30, delete "and tables" from the title of Appendix 2 since this appendix has only figures. | Appendix has been removed |
| p. 31, in the caption for Figure B1, replace "Fig.s" with "Figs." | Appendix has been removed |
| p. 34, Figure E1, for the top right part of the Figure (all avalanches for Switzerland), | Appendix has been removed |

add "(SWI)" after "all avalanches" to be consistent with the other headers. Also, add the percent number above the bar for size 1 avalanches under Low to match the other graphs in this Figure.

[revised manuscript text omitted]

²⁵removed: *where the snowpack is shallow, close to ridgelines, in bowls*

²⁶removed: contrast, in the CMAH ,

²⁷removed: ease of finding evidence of

²⁸removed: (Statham et al., 2018a)

²⁹removed: *isolated, specific* and *widespread*

³⁰removed: (Clark and Haegeli, 2018)

³¹removed: EAWS

³²removed: ,

³³removed: , are work

³⁴removed: but

³⁵removed: And

(2) Which combination of the actual value of the three factors [..³⁶ ]best describes the various danger levels? We present a methodology to generate data-driven stability distributions and to obtain class intervals describing the frequency of a given [..³⁷ ]snowpack stability class. Finally, we will compare the findings with currently used definitions in avalanche forecasting, as EADS and CMAH, and make recommendations for improvements towards more consistent usage of the danger scale.

**2  Data**

All the data described below were recorded for the purpose of operational avalanche forecasting in Norway (NOR; Norwegian Water Resources and Energy Directorate NVE) or Switzerland (SWI; WSL Institute for Snow and Avalanche Research SLF). In the vast majority, these observations were provided by specifically trained observers, belonging to the observer network of either the Norwegian or the Swiss avalanche warning service.

[..³⁸ ]For the analysis, we rely primarily on the Swiss data using the Norwegian data for comparison and validation. Nevertheless, we will occasionally present results for Swiss and Norwegian data side by side.

**2.1  Avalanche danger level**

[revised manuscript text omitted]

To enhance the quality of the data, we filtered observations, which we believe may indicate errors in the local estimate of the danger level or of avalanche size. To this end, we calculated the avalanche activity index (AAI,  Schweizer et al., 1998), a dimensionless index summing up avalanches according to their size with weights of 0.01, 0.1, 1, and 10 for avalanche sizes 1 to 4, respectively. We did not assign weights to the trigger type (natural, human-triggered). For NOR, where the number
* * *
[61]removed: 698

[62]removed: 682

[63]removed: NOR and SWI

[64]removed: SWI, observers reported the number of avalanches of a given size. In all reporting forms, information about the wetness and trigger type could be provided. In

of observed avalanches is described categorically, we assigned numbers as follows: one = 1, few (2-5) = 3, several (6-10) = 8, numerous (≥11) = 12. For each country, we then rank-ordered the avalanche data and the lowest 2.5% of the days and regions with 2-Moderate, 3-Considerable and 4-High, and the top 2.5% of the days and regions with 1-Low, 2-Moderate or 3-Considerable were considered to represent errors in the local estimate of the danger level or of avalanche size. These

5    potentially erroneous data were removed.

[revised manuscript text omitted]

[78]removed: *poor*, *poor-fair*, *fair* and *good*.It is of note, that ECT class *poor* also includes the weakest ECT results , which may be associated with *very poor* stability. To obtain the lowest RB or ECT stability class at each location, we proceeded as follows: If the depth of a weak layer failure was less than 5 cm below the snow surface

[79]removed: *good*

[80]removed: between 6 and

[81]removed: , we increased the stability rating by one step (e. g. from *very poor* to *poor*). If several failure planes were detected in a single stability test, the most unstable stability

[82]removed: snow
* * *
butions, we generated stability distributions by random sampling from the entire population of stability tests at a given danger level. Thus, we applied bootstrap sampling (Efron, 1979) and proceeded as follows (see also Fig. 2[..[83] ]a and b):

[revised manuscript text omitted]

* * *
[220]removed: considerably

[221]removed: ,

[222]removed: monotonically

[223]removed: more than 75

[224]removed: This proportion was higher in SWI (78%; Fig. **??**d) than in NOR (59%; Fig. **??**c).

[225]removed: weaker

[226]removed: p

[227]removed: than

[228]removed: p

[..[229] ]Note that we did not explore days with no avalanches as we were interested in the size of avalanches, not their frequency. The frequency component is addressed using the frequency of locations with [..[230] ]*very poor* stability as a proxy.

[revised manuscript text omitted]

*Competing interests.*  No competing interests.

20  *Acknowledgements.*  We thank the two reviewers Simon Horton and Karl Birkeland for their detailed and very helpful feedback, which greatly helped to improve this manuscript.
* * *
450removed: comparably little discriminating power at

451removed: (

452removed: an

453removed: spatial distribution of *very poor*

454removed: , like

[revised manuscript text omitted]

---

## Author Response (AR2)

**Response to reviewer comments**

Dear editor, dear reviewers,

thank you very much for reviewing our revised manuscript, and providing us with the opportunity to address the comments. Please find below a point-by-point reply (in blue) to the issues raised by reviewer Simon Horton *(in italics)*. Please also refer to the track-changes version of the revised manuscript.

**General comments**

*The authors have substantially improved the manuscript by addressing my main comment about the presentation of the results. The new structure addresses the objective of the paper in a more logical and intuitive way by directly addressing each of the key factors contributing to avalanche danger and then combining them into a practical look-up table. They have also addressed my other comments appropriately. My main remaining concern is the results section on bootstrap sampling is still slightly confusing with figures that don't necessarily support the main arguments in a clear way (see comment bellow). I also have some other minor comments on the revised manuscript below.*

**Specific comments**

– *Sect. 4.1.2: I appreciate how this section and Fig. 4 present the frequency of very poor stability in a clear way. I would consider modifying the section title and adding an introductory sentence to clarify that these frequencies come from sampling (e.g. "sampled" frequency of very poor stability). The example of what these frequencies mean in terms of 25 RB tests is a nice way to link these classes to practical field applications.* - We now provide some additional information regarding the sampling in the introductory sentence (p.10 l.7-8).

– *proportion vs. frequency: Please check the usage of these terms which are mixed throughout the paper. When discussing the key factor "frequency of triggering spots" it would make sense to use the term "frequency" (e.g. axis labels in Fig. 4, Fig. 7a, etc.). There is also a mix of reporting frequencies as a decimals and percentages (e.g. 0.04 and 4%). Using standard formatting and labelling for your derived frequencies would be helpful.* - Thank you for pointing this out. We have changed the formatting to percentages throughout the manuscript, including equations 1 and 2. Where it was clear that a reference was made to the frequency of triggering spots, we adjusted accordingly. In all other cases, we did not make further changes in this regard.

- *Fig. 4: Caption should be more detailed and explain how these proportions come from a sampling method. This will help explain how this figure differs from the data presented in Fig. 3. Also, this figure could be cited at the start of the paragraph about correlations between frequency classes and danger level (p11 l3).* - We now provide more detail in the caption to Fig. 4.

- *Table 2: Please explain what the interval means in the caption (i.e. proportion of very poor stability).* - We have added this information in the caption (Tab. 2).

- *Fig. 6: The percentages below the danger matrix are only explained in the text, but not in the figure or caption.* - We expanded the caption of Fig. 6. We now explain the percentages in the caption as well. Furthermore, and for each sub-figure, we added a sentence explaining the matrix workflow.

- *Fig. 6/7: This new figure layout links the lookup table and raw data in a much clearer way. Thanks.*

- *Sect 4.6: The section on bootstrap sampling is still rather complex and difficult to understand. I find lines 1-5 on page 18 and its links to Fig. 8 particularly challenging to understand. First do the results you describe in text refer to Fig. 8c as written or Fig. 8b? I'm not sure where I can see these values in Fig. 8. How does describing the percentage of samples with very poor and good stability relate to the typical distributions for each danger level? Are we supposed to compare these percentages with Fig. 2 or Fig. 9 to see that 11% of very poor results is within a typical range for 3-Considerable danger? Also, why is a threshold between few and several of 8% used here when elsewhere the threshold is 4%? Fig. 9 is also rather complex and difficult to interpret. Is the purpose to compare bootstrapping with n = 10 and n = 25 or is it to compare sampled and observed distributions? Based on the text in Sect 4.6.1 it seems to be about agreement between bootstrapped samples and field observations. If so, is it still necessary to have a 6-panel figure? While these extra results were appropriate in the Appendix their purpose in the main body of the results is less clear. Also, what is the significance of sentence in the caption about the distributions between Mod and Con danger for good stability n = 10? If this is important to validating the bootstrapping method it should be explained in the results. Overall I think this section and Fig 8 and 9 could still be presented in a clearer way to justify the bootstrapping method.* - We agree that the clarity of this section, and here particularly of Sect. 4.6.1 should be improved. Regarding the two figures (Fig. 8 and 9), and after careful consideration, we kept the two Figures in the main part of the manuscript. Moving these figures to the appendix would not have facilitated the understanding of this section. However, to improve the clarity of this section, we made several changes throughout Sect. 4.6.1: We now describe the aim of the section (p.17 l.22-24). We split the section into two parts - one on the influence of the sample size (p.17 l.26 - p.18 l.7), and one on plausibility of sampling (p.18 l.9-23). Furthermore, in the text, we point the reader more clearly towards specific results shown in the figures. We hope, these changes will increase the clarity of this Section.

[revised manuscript text omitted]

---

## Author Response (AR3)

Dear Editor, dear Editorial Team

thank you for accepting our manuscript for publication.

Kind regards,

Frank Techel – on behalf of all authors